# SQLENS: An End-to-End Framework for Error Detection and Correction in Text-to-SQL

**Yue Gong**[○]    **Chuan Lei**[◇∗]  **Xiao Qin**[⋆]   **Kapil Vaidya**[†]

**Balakrishnan Narayanaswamy**[○]    **Tim Kraska**[○,□]

[○]*Amazon Web Services*   [◇]*Oracle Corporation*   [⋆]*Snowflake Inc.*
[†]*Parallel AI*   [□]*Massachusetts Institute of Technology*

{yuegongy, muralibn, timkraska}@amazon.com,
chuan.lei@oracle.com, xiao.qin@snowflake.com, kapilvaidya24@gmail.com

## Abstract

Text-to-SQL systems translate natural language (NL) questions into SQL queries, enabling non-technical users to interact with structured data. While large language models (LLMs) have shown promising results on the text-to-SQL task, they often produce semantically incorrect yet syntactically valid queries, with limited insight into their reliability. We propose SQLENS, an end-to-end framework for fine-grained detection and correction of semantic errors in LLM-generated SQL. SQLENS integrates error signals from both the underlying database and the LLM to identify potential semantic errors within SQL clauses. It further leverages these signals to guide query correction. Empirical results on two public benchmarks show that SQLENS outperforms the best LLM-based self-evaluation method by 25.78% in F1 for error detection, and improves execution accuracy of out-of-the-box text-to-SQL systems by up to 20%.

## 1 Introduction

Text-to-SQL systems translate natural language (NL) questions into a SQL query, enabling non-technical users to query relational databases and extract insights [32]. Large Language Models (LLMs) have significantly advanced this task, with recent methods [44, 34, 29, 16] achieving promising results on public benchmarks such as BIRD [25] and Spider [55]. Such LLM-based text-to-SQL solutions have been adopted by data platforms like AWS [4], Databricks [12], Snowflake [41], etc.

Despite these advances, LLM-generated queries remain error-prone [14]. The best-performing method on the BIRD leaderboard [5] only achieves an execution accuracy of around 75% on the dev set, still producing more than 400 incorrect SQL queries out of 1534 NL questions. While considerable effort has been devoted to improving the retrieval and generation stages in LLM-based text-to-SQL systems [44, 34, 29], such as schema linking, Chain-of-Thought (CoT) prompting, task decomposition, and the inclusion of cell values, these approaches are still insufficient to eliminate errors entirely, as LLMs do not always adhere to the provided instructions.

A major gap in text-to-SQL systems is the lack of fine-grained, explainable error detection–especially for semantic errors, where queries execute but return incorrect results. Detecting such errors requires understanding both query logic and database structure, which most existing methods fail to capture [33, 48, 23]. They generate SQL queries without estimating quality or confidence, making it difficult for users to debug [42]. In this paper, we address three challenges in error detection and correction:

---

[∗]Work done at AWS by Chuan Lei, Xiao Qin, and Kapil Vaidya.

39th Conference on Neural Information Processing Systems (NeurIPS 2025).

**Challenge 1: identify semantic errors at the clause level.** Semantic errors often arise from various causes, from inconsistencies with external knowledge to the LLM's misunderstanding of the database schema. Identifying them is challenging due to NL ambiguity and the complexity of SQL queries, data, and schema [47, 18], and requires joint reasoning over all these sources.

**Challenge 2: predict the correctness of a SQL query with noisy error signals.** Error signals are noisy proxies for true semantic errors. Each signal evaluates the semantic correctness of the generated SQL from a specific perspective, potentially missing the full context. The challenge is to aggregate the decisions from multiple noisy signals to determine whether a SQL query is semantically incorrect.

**Challenge 3: fix SQL queries without correctness oracle.** Correcting SQL queries is difficult because it must be done cautiously to avoid breaking queries that may already be correct, as no correctness oracle is assumed. Additionally, the order in which errors are fixed is crucial, as errors can be interdependent, and addressing the root cause can resolve many related issues.

**State-of-the-art approaches.** Existing text-to-SQL methods often rely on LLM self-reflection for error detection and correction [33, 23, 34], but suffers from self-preference bias [31, 52], resulting in low recall. RL-based approaches are emerging, using rewards based on execution accuracy, syntactic correctness, or LLM evaluations [36, 27], but still lack the granularity needed for precise error identification. These methods typically offer only a single confidence score for the full SQL query, without pinpointing specific erroneous clauses or explaining the cause—limiting debuggability and reducing trust in text-to-SQL assistants [6].

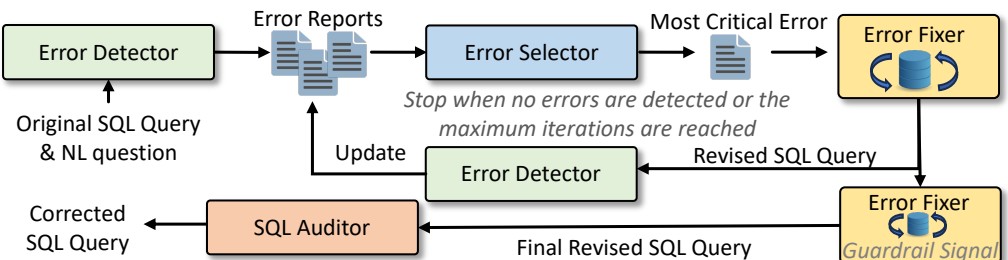

Figure 1: An overview of SQLENS.

**SQLENS overview.** We introduce SQLENS, an end-to-end framework that leverages diverse error signals from both the database and LLM to detect and correct semantic SQL errors. As shown in Figure 1, SQLENS parses the SQL query into an Abstract Syntax Tree (AST) and uses its `Error Detector` to collect noisy signals for identifying potential errors (*Challenge 1*). To handle noisy signals, SQLENS uses weak supervision to combine them into probabilistic labels–without needing ground truth–and trains a classifier to predict semantic correctness and generate detailed error reports (*Challenge 2*). SQLENS's `Error Selector` then decomposes the SQL correction task and guides the LLM through iterative fixes (`Error Fixer`) based on prioritized errors (*Challenge 3*), and the `SQL Auditor` selects the better query between the original and revised versions.

**Contributions.** Our contributions can be summarized as follows: (1) we present SQLENS that holistically exploits fine-grained error signals for predicting the semantic correctness of SQL queries and correcting erroneous queries using these signals; (2) SQLENS leverages both LLM and database signals to diagnose SQL queries and employs a weak supervision framework to aggregates noisy error signals for estimating semantic correctness; (3) we design a sequential error correction strategy that guides the LLM to fix SQL queries step-by-step, prioritizing the most critical errors to minimize cascading mistakes; and (4) experimental results on BIRD [25] and Spider [55] show that SQLENS improves semantic error detection by 25.78% in F1 score over the best LLM self-evaluation method and boosts execution execution accuracy by an average of 3% over existing text-to-SQL solutions.

## 2   Related Work

**Text-to-SQL.** Large language models (LLMs) have demonstrated significant capabilities in natural language understanding as the model scale increases. LLM-based text-to-SQL systems, such as

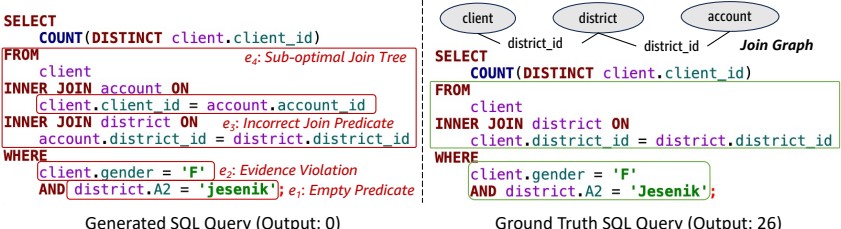

**Question (𝒬)** How many clients opened their accounts in Jesenik branch were women?
**External Knowledge (𝒦):** A2 has region names; female refers to gender = 'F'

Figure 2: A running example from BIRD, where each client is guaranteed to have an account.

DIN-SQL [33] and MAC-SQL [48], decompose the task into sub-tasks using multi-agent frameworks for effective execution. Other approaches like MCS-SQL [23], CHESS [44] and Chase-SQL [34] follow similar workflows—retrieving relevant context, selecting schema, and synthesizes SQL queries. Recently RL-based reasoning models for text-to-SQL have been explored, where rewards are typically based on execution accuracy, syntactic correctness, or LLM-based self-evaluation [36, 27]. Our proposed error detection can be naturally integrated into this framework as reward functions, providing more semantically meaningful supervision to guide and enhance model reasoning.

**Error detection and correction in text-to-SQL.** Early approaches [8, 53, 54, 56] focus primarily on binary correctness classification, offering limited insight into the nature of the errors. LLM-based text-to-SQL systems [33, 48, 35, 23] often rely on either SQL execution feedback or LLM self-reflection to judge correctness. In contrast, SQLLENS produces interpretable, clause-level error signals that can guide both user debugging and downstream learning frameworks. LLM self-evaluation techniques [20, 17] assess response quality, while Tian et al. [45] and Self-RAG [1] improve confidence estimation and factual accuracy through fine-tuning and retrieval-based self-critique. However, these are built for general tasks and do not tackle text-to-SQL's specific challenges.

## 3 Methodology

### 3.1 Problem Formulation

In this paper, a table $T$ consists of a set of columns $\mathcal{C} = \{C_1, C_2, ..., C_n\}$. A join relationship $J$ between two tables $T_i$ and $T_j$ is based on common attributes (i.e., joinable columns $T_i.C_m$ and $T_j.C_n$, respectively). A database instance $\mathcal{D} = \{(T_1, T_2, \ldots, T_n), \mathcal{J}\}$ comprises a set of tables and a set of join relationships $\mathcal{J}$ between these tables.

**Definition 1 (Text-to-SQL Algorithm)** *A text-to-SQL algorithm $f$ takes as input a natural language question $\mathcal{Q}$, a database instance $\mathcal{D}$, and optionally external knowledge $\mathcal{K}$, and generates a SQL query $q = f(\mathcal{Q}, \mathcal{D}, \mathcal{K})$.*

Figure 2 presents an example using a question from the BIRD benchmark, asking for the number of female clients who opened accounts at the Jesenik branch. The benchmark also provides external knowledge, such as annotations on a column name and a cell value. The predicted SQL query is generated by a text-to-SQL algorithm to answer the question.

**Definition 2 (Semantic Error)** *A semantic error $e$ results in the SQL query $q$ failing to correctly answer the natural language query $\mathcal{Q}$. Formally,*

$$do(e) \Rightarrow \mathcal{O}(q, \mathcal{D}) \neq \mathcal{O}(\mathcal{Q}, \mathcal{D})$$

The operation $do(e)$ denotes an intervention in the generation of $q$ due to $e$, causing a mismatch between the observed output $\mathcal{O}(q, \mathcal{D})$ and the expected correct output $\mathcal{O}(\mathcal{Q}, \mathcal{D})$, thereby identifying $e$ as the semantic error.

For example, the generated SQL query in Figure 2 is semantically incorrect with an output of 0, whereas the correct output is 26 based on the ground truth SQL query. First, the query incorrectly

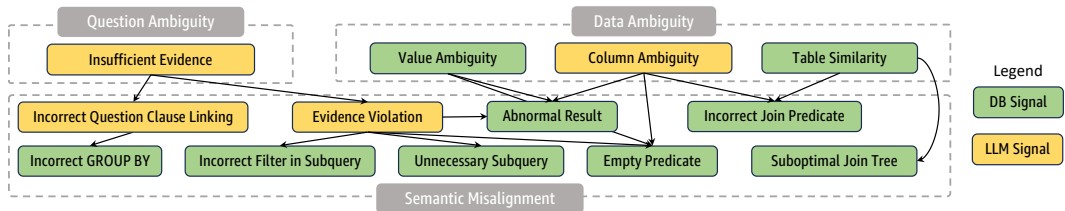

Figure 3: The causal graph of semantic errors and signals

uses `jesenik` in the predicate, leading to the empty result. Secondly, the query violates the evidence specified in the external knowledge, using `gender='Female'` instead of `gender='F'`. Even with the correct predicates, the SQL query would still produce an incorrect result of 23 due to an incorrect join predicate and a suboptimal join path. Specifically, there is no valid join path between the `client` and `account` tables, indicating the join predicate `client.client_id = account.account_id` in the query is hallucinated. Moreover, the `account` table involved in the join path is redundant as the `client` and `district` tables can be directly joined based on the database.

**Problem 3.1 (Semantic Error Detection)** *Given a natural language question $\mathcal{Q}$, a database instance $\mathcal{D}$, optionally external knowledge $\mathcal{K}$, and the output SQL query $q = f(\mathcal{Q}, \mathcal{D}, \mathcal{K})$ generated by a text-to-SQL algorithm $f$, the task is to determine if $q$ is semantically correct and, if not, identify a set of potential semantic errors $\mathcal{E}$ in $q$.*

For a semantically incorrect SQL query, our goal is to fix it to correctly answer the given NL question.

**Problem 3.2 (Semantic Error Correction)** *Given a semantically incorrect SQL query $q$, the task is to correct $q$ such that the corrected query $q'$, accurately answers the natural language question $\mathcal{Q}$, such that $\mathcal{O}(\mathcal{Q}, \mathcal{D}) = \mathcal{O}(q', \mathcal{D})$.*

### 3.2 SQLENS Error Detection

Accurately identifying semantic errors can be inherently difficult due to the complexity of SQL queries, as well as the ambiguities present in natural language (NL) queries, data, and database schemas. Our insight is that many semantic errors in LLM-generated queries exhibit common patterns that can be detected through carefully crafted signals. Building on observations and insights from recent text-to-SQL studies [48, 23, 35], we categorize common semantic errors as follows.

1. Question ambiguity. The user's questions may inherently contain ambiguities and be interpreted in multiple ways. For example, if a user asks "*What were the total sales last quarter?*" in a database where the `sales` table includes both `gross_sales` and `net_sales` columns, either column could be chosen to answer the question.

2. Data ambiguity. In real-world databases, multiple tables or columns with similar or identical names can arise due to data integration, versioning, table transformations, and other factors, leading to ambiguities. For example, if a user asks "*What are the average salaries by department?*", but the database contains both a `dept` table and a `department` table, selecting the wrong table would result in incorrect query results.

3. Semantic misalignment. Even when NL questions and databases are unambiguous, a semantic gap can still cause mismatches between the generated SQL and the NL question. For example, as shown in Figure 2, an incorrect join predicate (`client.client_id = account.account_id`) occurs due to the LLM misinterpreting join relationships in the financial database.

An *error signal* acts as a proxy for detecting semantic errors in SQL queries. SQLENS's `Error Detector` combines signals from both the database and LLM to detect these errors. Figure 3 illustrates how each signal in SQLENS corresponds to common types of semantic errors.

**Database-based error signals.** Database-based signals are designed to identify semantic misalignment and inherent ambiguity within the data. These signals draw inspiration from diverse SQL workloads including real-life scenarios and benchmarks such as TPC-DS [30], Redset [46], BIRD [25], etc. These signals can be reliably and efficiently obtained without LLMs by: (1) extracting

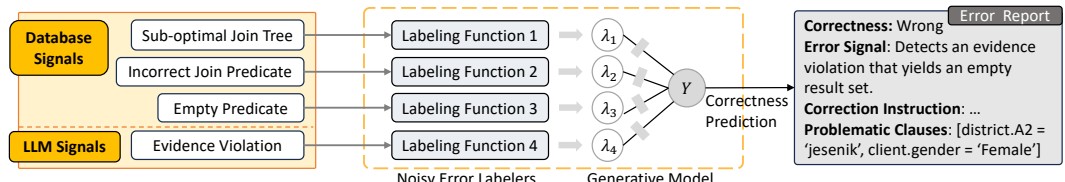

Figure 4: Signal aggregation using weak supervision.

information from the query plan (e.g., *Empty Predicate*), (2) checking (meta-)data information from the underlying database (e.g., *Suboptimal Join Tree, Value Ambiguity*), or (3) leveraging general heuristics derived from the above query workloads (e.g., *Unnecessary Subquery*). In Appendix B, we present examples of incorrect and correct SQL query pairs based on NL questions from real-world applications, highlighting the specific SQL clauses each signal targets.

**LLM-based error signals.** For semantic errors due to insufficient evidence or LLM hallucination, we need to consider both the NL question and the SQL query simultaneously and understand the LLM's reasoning process. Therefore, SQLENS introduces LLM-based error signals that dig into question ambiguity and the LLM's reasoning process [50]. All LLM-based signals are obtained from a single LLM call, significantly reducing the cost and latency compared to multiple individual calls. Detailed descriptions of the error signals, along with illustrative examples, are provided in Appendix C. The corresponding prompts used to collect these signals are included in Appendix D.

Effective error detection in text-to-SQL requires integrating both database and LLM-based signals: the former excel at identifying structural or value errors, while the latter capture logical inconsistencies requiring nuanced interpretation of the NL question and query. Since these signals are inherently noisy, SQLENS avoids relying on any single source. Instead, it employs a learning-based approach (Section 3.3) to aggregate complementary signals, combining database analysis with LLM reasoning. This makes SQLENS a general and extensible framework for robust error detection and correction.

### 3.3 Aggregating All Signals Using Weak Supervision

The signals that we collect are inherently noisy—each may incorrectly flag correct SQL clauses or overlook actual errors. To mitigate this, we adopt a weak supervision framework (shown in Figure 4) that aggregates these diverse signals, treating each as a labeling function (LF) that provides partial and noisy supervision. Inspired by prior work [38, 15], this approach not only combines the signals but also learns their accuracies and correlations to better approximate ground truth. In SQLENS, each error signal $s$ acts as a LF, identifying potential issues in a SQL query $q$. Formally,

$$\mathcal{I} = \lambda_s(q) = \begin{cases} 1 & \text{if } |s(q)| > 0 \\ -1 & \text{if } |s(q)| = 0 \end{cases}$$

where $\mathcal{I} = 1$ indicates that the error signal $s$ has detected at least one problematic SQL clause (i.e., likely incorrect), and $\mathcal{I} = -1$ indicates that no issues are found. However, relying solely on negative signals limits coverage. To improve recall, we introduce three positive labelers that assign $\mathcal{I} = 0$ (i.e., likely correct) when no issues are detected by: (1) $\lambda_{all}$ labels a SQL query as correct if no error signals are detected; (2) $\lambda_{db}$ labels a SQL query as correct if no database-based signals are detected; and (3) $\lambda_{llm}$ labels a SQL query as correct if no LLM-based signals are detected.

Each query $q$ is thus represented by a decision vector $\Lambda_q = \langle \lambda_{s_1}(q), ..., \lambda_{s_n}(q), \lambda_{all}(q), \lambda_{db}(q), \lambda_{llm}(q) \rangle$. We use a generative model to estimate the join distribution $p(\Lambda, Y)$, where $Y$ is the unobserved true label, by optimizing:

$$\theta_{opt} = \arg \min_{\theta} - \log \sum_{Y} p_{\theta}(\Lambda, Y)$$

The resulting probabilistic labels $p(\Lambda, Y)$ reflect the estimated correctness of each SQL query, which we use to train a classifier for downstream semantic error detection.

```
{
    "signal description": "The SQL query uses more tables than necessary in the join
    ↪  clauses, which may lead to potential errors.",
    "correction instruction": "Review and revise the SQL query to include only the
    ↪  essential tables in the join clauses.",
    "problematic clauses": {
        "tables used in the JOIN clauses": ["client", "account", "district"],
        "optimal set of tables to join": ["client", "district"]
    },
    "confidence": "high"
}
```
Figure 5: Example error report for *Suboptimal Join Tree*

### 3.4 SQLENS Error Correction

As discussed in Section 1, LLM-based text-to-SQL methods [33, 9] often rely on self-reflection without grounding their corrections. This leads to two issues: (1) hallucinations due to generation bias [31, 52] and limited factual grounding [39], and (2) degradation of originally correct queries when relying on a single, unreliable correctness estimate. Other methods similarly assume that all SQL queries are incorrect [11, 2], ignoring the risk of over-correction.

**SQLENS error report.** In SQLENS, an error signal is associated with SQL clauses and provides a detailed error report. An error report consists of (1) *signal description* which explains the error signal and the conditions that trigger it; (2) *example(s)* (optional) to clarify the meaning of the signal; (3) *correction instruction* for correcting the SQL query; (4) *problematic clauses* identified as potential sources of error. Such error report offers rich context for the identified errors in SQL queries; and (5) *confidence* bucketized into high, medium, and low based on the weight assigned to each labeler in error detection. Figure 5 presents an example of an error report generated by the *Suboptimal Join Tree* signal for the SQL query shown in Figure 2.

**SQLENS correction strategy.** A straightforward approach of using error signals for SQL query correction is to concatenate all the detected error information and ask the LLM to fix all errors at once. However, providing too much information can overwhelm the LLM, causing it to lose focus [26]. To address this, SQLENS decomposes the SQL correction task, guiding the LLM to fix the query step-by-step. SQLENS prioritizes fixing the most critical error in each iteration, allowing the LLM to focus on resolving one issue at a time, reducing the risk of distraction [21, 49, 51]. By addressing the most critical error first, the total number of errors can also be reduced more quickly, as dependent errors can be automatically resolved if the root cause is fixed. This iterative process continues until no error signals remain or the maximum number of iterations is reached. Algorithm 1 in Appendix E presents the pseudocode for the error correction. The correction prompt is provided in Appendix D.8.

• **Error Selector.** Given a SQL query, schema, and a ranked list of error reports, the `Error Selector` uses an LLM to prioritize which error to fix first, refining the ranking based on relevance to the original query [43, 28, 37]. The highest-priority error is then passed to the `Error Fixer`.

• **Error Fixer.** The `Error Fixer` takes the query and a selected error report, applies a targeted correction using contextual signals (Algorithm 1), and validates the revised query using a SQL parser. If syntax issues arise, the fixer uses the parser feedback to iteratively resolve them. Corrected queries are re-evaluated by the `Error Detector`; the process repeats until no errors remain or a maximum number of iterations is reached. Fixed signals are removed to prevent redundant corrections.

• **SQL Auditor.** To avoid overlooking persistent errors, a high-precision signal can be designated as a guardrail to trigger final adjustments. To prevent degradation of originally correct queries, the final revised query and the original are both evaluated by an LLM-based `SQL Auditor`, which selects the version that best aligns with the input question.

## 4 Experimental Evaluation

### 4.1 Experimental Setup

**Datasets.** We evaluate on the dev sets of two standard text-to-SQL benchmarks: BIRD [25] and Spider [55]. SQL queries for evaluation were generated using four state-of-the-art text-to-SQL

Table 1: End-to-end accuracy improvement on BIRD.

| | Method | $\Delta$ Acc.($N_{\text{net}}$) | $\Delta$ Acc.($N_{\text{fix}}$) | $N_{\text{net}}$ | $N_{\text{fix}}$ | $N_{\text{break}}$ |
|---|---|---|---|---|---|---|
| Vanilla | Self-Reflection | 59.07 (+0.00%) | 60.6(+1.53%) | 1 | 22 | 21 |
| | SQLENS w. Fix-ALL | 61.15 (+2.08%) | 62.34 (+3.27%) | 30 | 47 | 17 |
| | SQLENS | **62.54 (+3.47%)** | **63.66(+4.59%)** | **50** | **66** | **16** |
| DIN-SQL | Self-Reflection | 54.63 (+15.14%) | 56.37(+16.88%) | 209 | 233 | 24 |
| | SQLENS w. Fix-ALL | 58.11 (+18.62%) | 59.71(+20.22%) | 257 | 279 | 22 |
| | SQLENS | **59.99 (+20.50%)** | **61.45 (+21.96%)** | **283** | **303** | **20** |
| MAC-SQL | Self-Reflection | 60.12 (+0.80%) | 62.63(+2.59%) | 12 | 39 | 27 |
| | SQLENS w. Fix-ALL | 62.43 (+2.39%) | 63.84(+3.80%) | 36 | 57 | 21 |
| | SQLENS | **64.03 (+3.99%)** | **65.23 (+5.19%)** | **60** | **78** | **18** |
| CHESS | Self-Reflection | 67.99 (+0.08%) | 69.42 (+1.51%) | 12 | 23 | **11** |
| | SQLENS w. Fix-ALL | 68.96 (+1.05%) | 70.21(+2.30%) | 16 | 35 | 19 |
| | SQLENS | **69.74 (+1.83%)** | **70.53(+2.62%)** | **28** | **40** | 12 |

systems with Claude 3.5 Sonnet as the backbone LLM: (1) a basic schema-aware prompt (Vanilla) listed in Appendix D.1, (2) DIN-SQL [33], (3) MAC-SQL [48], and (4) CHESS [44]. Semantic correctness was determined by comparing execution results of generated queries against gold queries.

**Evaluation metrics.** We evaluate SQLENS on two tasks: (1) *end-to-end correction*, measuring execution accuracy gains on benchmarks; and (2) *semantic error detection*, framed as binary classification. For correction, in addition to the execution accuracy gain $\Delta$ Acc., we also report $N_{\text{fix}}$ (incorrect $\rightarrow$ correct), $N_{\text{break}}$ (correct $\rightarrow$ incorrect), and the net improvement $N_{\text{net}} = N_{\text{fix}} - N_{\text{break}}$. For error detection, we treat semantically incorrect queries as the positive class and report accuracy, AUC, and precision/recall/F1, over 5-fold cross-validation.

**End-to-end correction baselines. (1) Self-Reflection**: A standard LLM prompt [33] where the model is given the schema, sample values, and original SQL query, and asked to debug without any explicit error information. **(2) SQLENS + Fix-ALL**: Provides the LLM with all error signals detected by SQLENS simultaneously, prompting it to correct the query based on the full set of error reports.

**Semantic error detection baselines. (1) LLM Self-Eval (Bool)**: Following Kadavath et al. [20], the LLM is asked whether the SQL query correctly answers the question (yes/no). **(2) LLM Self-Eval (Prob)**: Based on Tian et al. [45], the LLM outputs a confidence score (0–1) for query correctness; scores above 0.5 are treated as correct. **(3) Supervised SQLENS**: A classifier trained on error signal outputs as features and gold correctness labels derived by comparing the generated SQL against the ground truth. We use AutoGluon [13] to automatically select the best-performing model. All methods, including SQLENS, use Claude 3.5 Sonnet as the backbone LLM.

**SQLENS Setup.** SQLENS is implemented in Python 3.9, with the weak supervision component built using Snorkel's LabelModel [40]. We configure SQLENS to perform up to 3 correction iterations.

## 4.2 Main Results

As shown in Table 1, SQLENS consistently improves end-to-end accuracy across all out-of-the-box text-to-SQL systems and outperforms both `Self-Reflection` and `Fix-ALL` on BIRD. In this realistic setting–where inputs include both correct and incorrect queries–SQLENS achieves the highest net improvement ($N_{\text{net}}$), fixing more queries and introducing fewer regressions. For example, with queries generated by the vanilla prompt, SQLENS corrects 49 and 20 more queries than `Self-Reflection` and `Fix-ALL`, respectively, yielding 3.47% and 2.08% higher accuracy. It improves DIN-SQL's performance by 20.5% and boosts CHESS from 67.91% to 69.74%. When assuming all inputs are incorrect–as done by other methods–SQLENS achieves an average gain of 4.13%. Notably, LLM self-reflection breaks the most correct queries, highlighting the need for grounded, signal-driven correction. The results on Spider are in Appendix F.1.

**Comparison with state-of-the-art correction frameworks.** To contextualize our improvements, we further compare SQLENS against recent error-correction systems: MAGIC [3], Self-Debug [10], and SQLFixAgent [7] on the BIRD dev set. As shown in Table 2, SQLENS achieves the highest execution accuracy post-correction (65.23%), surpassing the strongest baseline (SQLFixAgent at 60.17%) by 5.06 points. Despite starting from a stronger base SQL generation method (MAC-SQL, 60.04% EX accuracy vs. 57.17% in prior work), SQLENS still achieves to the largest relative improvement (+5.19%), demonstrating its robustness and scalability even when built on high-quality initial queries.

Table 2: Comparison with recent SQL correction frameworks on the BIRD dev set. Each framework is evaluated under the SQL generation method that yields its best reported results.

| Correction Framework | SQL Generation Method | EX Acc. (Initial → Corrected) |
|---|---|---|
| MAGIC [3] | DIN-SQL [33] | 56.52 → 59.13 (+2.61) |
| Self-Debug [10] | DIN-SQL [33] | 56.52 → 57.24 (+0.72) |
| SQLFixAgent [7] | CodeS-7b [24] | 57.17 → 60.17 (+3.00) |
| SQLENS (Ours) | MAC-SQL [48] | **60.04 → 65.23 (+5.19)** |

### 4.3 Ablation Studies

**Error detection performance analysis.** We evaluate SQLENS's effectiveness in predicting semantic correctness. Results on BIRD are shown in Table 3, with additional results on Spider provided in Appendix F.2.

On BIRD, SQLENS outperforms all baselines in Accuracy, Recall, and F1, demonstrating superior ability to identify erroneous SQL queries. For DIN-SQL, SQLENS achieves an F1 of 78.88, 21.45 points higher than the best LLM self-evaluation baseline (57.43). Although LLM Self-Eval. (Prob) attains the highest precision on MAC-SQL, it suffers from the lowest recall.

When aggregating signals, the supervised version of SQLENS achieves higher accuracy, AUC, and precision, benefiting from access to gold labels during training to better assess signal reliability. In contrast, the weak supervision variant yields higher recall and F1 by leveraging agreement among signals without relying on ground truth. While this leads to slightly lower precision and accuracy, it enables broader error coverage by capturing diverse aspects of SQL errors.

We further analyze the impact of SQL Auditor and the guardrail signal on SQLENS error correction through an ablation study on BIRD, as shown in Table 4.

**Impact of SQL Auditor.** SQL Auditor helps reduce the number of SQL queries broken during the correction process. However, this comes at the expense of fixing fewer queries overall. The decision to enable SQL Auditor depends on the desired objective: whether to prioritize conservative corrections with minimal regressions or more aggressive corrections aimed at maximizing the total number of fixed queries.

Table 4: Effectiveness of SQL Auditor and guardrail signal on BIRD.

| | Method | $N_{net}$ | $N_{fix}$ | $N_{break}$ |
|---|---|---|---|---|
| Vanilla | SQLENS | 50 | 66 | 16 |
| | w/o SQLAuditor | 53 | 72 | 19 |
| | w/o Guardrail+Auditor | 48 | 69 | 21 |
| DIN-SQL | SQLENS | 283 | 303 | 20 |
| | w/o SQLAuditor | 288 | 308 | 20 |
| | w/o Guardrail+Auditor | 281 | 304 | 23 |
| MAC-SQL | SQLENS | 60 | 78 | 18 |
| | w/o SQLAuditor | 57 | 78 | 21 |
| | w/o Guardrail+Auditor | 46 | 69 | 23 |
| CHESS | SQLENS | 28 | 40 | 12 |
| | w/o SQLAuditor | 26 | 40 | 14 |
| | w/o Guardrail+Auditor | 24 | 38 | 14 |

**Impact of guardrail signal.** To isolate the effect of the guardrail signal, we compare the results with and without it before running SQL Auditor. As shown in Table 4, the use of the guardrail signal increases the net improvements by raising the total number of fixes while reducing the total number of regressions.

### 4.4 Effectiveness of Individual Signals

We conduct a microbenchmark on BIRD to evaluate the effectiveness of each individual signal in both error detection and correction.

Table 3: Effectiveness of SQLENS error detection on BIRD.

| | Method | Acc. | AUC | Precision | Recall | F1 |
|---|---|---|---|---|---|---|
| Vanilla | LLM Self-Eval. (Bool) | 60.53 (±2.30) | ✗ | 57.70 (±18.90) | 10.02 (±4.54) | 16.97 (±7.35) |
| | LLM Self-Eval. (Prob) | 59.76 (±1.10) | 64.23 (±2.98) | 64.22 (±22.97) | 2.72 (±1.89) | 5.17 (±3.49) |
| | Supervised SQLENS | 66.58$^\dagger$ (±2.56) | 65.12$^\dagger$ (±3.88) | 71.83$^\dagger$ (±9.55) | 31.77 (±7.38) | 43.32 (±6.47) |
| | SQLENS | 64.63 (±1.97) | 61.90 (±2.18) | 58.11 (±2.80) | 48.74$^\dagger$ (±4.31) | 52.94$^\dagger$ (±3.24) |
| DIN-SQL | LLM Self-Eval. (Bool) | 61.52 (±1.20) | ✗ | 86.83 (±2.18) | 42.99 (±2.72) | 57.43 (±2.20) |
| | LLM Self-Eval. (Prob) | 49.57 (±1.56) | 73.01 (±0.86) | 92.27$^\dagger$ (±5.05) | 17.84 (±2.19) | 29.83 (±3.09) |
| | Supervised SQLENS | 76.96$^\dagger$ (±2.10) | 83.55$^\dagger$ (±2.33) | 85.90 (±2.14) | 74.13 (±3.27) | 79.53$^\dagger$ (±2.14) |
| | SQLENS | 75.29 (±2.33) | 81.49 (±1.74) | 81.64 (±2.17) | 76.41$^\dagger$ (±3.50) | 78.88 (±2.19) |
| MAC-SQL | LLM Self-Eval. (Bool) | 61.50 (±1.47) | ✗ | 65.51 (±14.72) | 7.83 (±2.16) | 13.94 (±3.69) |
| | LLM Self-Eval. (Prob) | 61.37 (±0.81) | 64.60 (±2.19) | 72.69$^\dagger$ (±12.19) | 5.16 (±1.53) | 9.61 (±2.76) |
| | Supervised SQLENS | 67.09 (±3.56) | 65.07$^\dagger$ (±4.30) | 66.19 (±10.83) | 38.11 (±2.90) | 48.16 (±4.29) |
| | SQLENS | 67.43$^\dagger$ (±4.38) | 64.27 (±4.42) | 63.27 (±8.64) | 45.10$^\dagger$ (±3.90) | 52.63$^\dagger$ (±5.62) |
| CHESS | LLM Self-Eval. (Bool) | 67.98 (±1.95) | ✗ | 50.89 (±10.76) | 15.71 (±3.96) | 23.81 (±5.22) |
| | LLM Self-Eval. (Prob) | 68.50 (±0.82) | 64.23$^\dagger$ (±1.65) | 61.87 (±13.09) | 4.90 (±1.63) | 9.03 (±2.87) |
| | Supervised SQLENS | 72.10$^\dagger$ (±1.27) | 62.95 (±3.23) | 72.43$^\dagger$ (±8.93) | 23.06 (±7.23) | 33.96 (±8.04) |
| | SQLENS | 69.35 (±1.60) | 63.34 (±2.90) | 52.54 (±2.91) | 44.69$^\dagger$ (±6.03) | 48.17$^\dagger$ (±4.33) |

✗ indicates that the classification is not threshold-based. $\dagger$ Top result; top two are bolded.

Table 5: Error detection using individual error signal.

| Signal Name | DIN-SQL | | | MAC-SQL | | | CHESS | | |
|---|---|---|---|---|---|---|---|---|---|
| **DB-based Error Signals** | Precision | Recall | $N_w$ | Precision | Recall | $N_w$ | Precision | Recall | $N_w$ |
| Abnormal Result | 99.68 | 37.01 | 309 | 100 | 6.66 | 40 | 98.48 | 13.27 | 65 |
| Empty Predicate | 96.07 | 46.83 | 391 | 75.81 | 7.82 | 47 | 81.25 | 10.61 | 52 |
| Incorrect Filter in Subquery | No queries detected | | | 76 | 3.16 | 19 | No queries detected | | |
| Incorrect GROUP BY | 68.75 | 5.27 | 44 | 66.67 | 3.0 | 18 | 50 | 0.2 | 1 |
| Incorrect Join Predicate | 100 | 0.12 | 1 | 92.86 | 2.16 | 13 | 100 | 0.41 | 2 |
| Suboptimal Join Tree | 73.86 | 15.57 | 130 | 62.24 | 10.15 | 61 | 55.13 | 8.78 | 43 |
| Table Similarity | 73.08 | 4.55 | 38 | 67.31 | 5.82 | 35 | 63.27 | 6.33 | 31 |
| Unnecessary Subquery | 92.31 | 1.44 | 12 | 62.77 | 10.15 | 61 | 100 | 0.2 | 1 |
| Value Ambiguity | 66.22 | 5.87 | 49 | 58.49 | 5.16 | 31 | 38.89 | 5.71 | 28 |
| **LLM-based Error Signals** | Precision | Recall | $N_w$ | Precision | Recall | $N_w$ | Precision | Recall | $N_w$ |
| Column Ambiguity | 88.69 | 17.84 | 149 | 75.0 | 3.49 | 21 | 50 | 4.69 | 23 |
| Evidence Violation | 91.24 | 14.97 | 125 | 87.1 | 4.49 | 27 | 42.86 | 1.22 | 6 |
| Insufficient Evidence | 82.4 | 12.34 | 103 | 62.07 | 3.0 | 18 | 41.03 | 3.27 | 16 |
| LLM Self-Check | 86.71 | 43.0 | 359 | 65.28 | 7.82 | 47 | 50.33 | 15.71 | 77 |
| Question Clause Linking | 87 | 12.34 | 103 | 60.71 | 5.66 | 34 | 53.49 | 4.69 | 23 |

Table 6: Error correction via each error signal on MAC-SQL.

| Signal Name | $N_{fix}$ | $N_{break}$ | $N_{net}$ |
|---|---|---|---|
| **DB-based Signals** | | | |
| Abnormal Result | 4 | 0 | 4 |
| Empty Predicate | 8 | 1 | 7 |
| Incorrect Filter in Subquery | 9 | 3 | 6 |
| Incorrect GROUP BY | 5 | 2 | 3 |
| Incorrect Join Predicate | 5 | 1 | 4 |
| Suboptimal Join Tree | 16 | 7 | 9 |
| Table Ambiguity | 2 | 1 | 1 |
| Unnecessary Subquery | 6 | 4 | 2 |
| Value Ambiguity | 4 | 1 | 3 |
| **LLM-based Signals** | | | |
| Column Ambiguity | 4 | 1 | 3 |
| Evidence Violation | 9 | 1 | 8 |
| LLM Self-Check | 10 | 3 | 7 |

**Individual signal in error detection.** Table 5 presents the performance of individual error signals on SQL queries generated by MAC-SQL, DIN-SQL, and CHESS on BIRD. For MAC-SQL, 13 of 14 signals exceed 60% precision. Notably, *Abnormal Result* achieves 100% precision in identifying 40 errors, and *Suboptimal Join Tree* detects the most errors, though at 62% precision. Signals like *Empty Predicate* and *Incorrect Join Predicate* perform consistently well across all systems.

In contrast, signals such as *Unnecessary Subquery* and *Insufficient Evidence* show lower precision and limited impact due to low coverage on the queries generated by MAC-SQL and CHESS. Among LLM-based signals, LLM Self-Check is the most reliable, consistent with results in Table 3.

The lower precision of *Suboptimal Join Tree* with MAC-SQL and CHESS stems from redundant joins that do not affect query results, though the suggested optimal trees often align with the gold query. While such joins may not cause semantic errors, they can harm execution efficiency. *Incorrect Join Predicate* consistently achieves over 90% precision. Overall, database-based signals provide better coverage and precision than LLM-based ones.

**Individual signal in error correction.** We evaluate the correction effectiveness of individual error signals using SQL queries generated by MAC-SQL. For each signal, we pass all flagged queries and their corresponding error reports to the relevant `Signal Fixer`. Results are shown in Table 6. Among database-based signals, *Suboptimal Join Tree* achieves the strongest impact with $N_{fix} = 16$ and a net gain of 9 corrected queries. *Empty Predicate* also performs well, fixing 8 and breaking only 1. For LLM-based signals, *Evidence Violation* yields the highest net gain, correcting 8 queries. All signals produce a positive net correction, highlighting the usefulness of SQLENS's error signals. We

also analyzed the fix sequences for SQLs generated by DIN-SQL. Among 415 queries with multiple detected errors, `Error Selector` most often chose *Empty Predicate* (248 cases) as the first fix, followed by *Evidence Violation* (98 cases) and *Column Ambiguity* (36 cases). Empty Predicate errors are typically addressed first because they are unambiguous and directly verifiable through execution results, making them strong anchors for initiating correction.

### 4.5 Token Cost Analysis

We report the detailed token consumption of SQLENS on the BIRD dataset. The four main stages– `Error Detector`, `Error Selector`, `Error Fixer`, and `SQL Auditor`–consume an average of 3,141, 2,345, 2,262, and 2,215 tokens per query, respectively. A substantial portion of these tokens (approximately 1,915 per component) comes from the database description, which contains the schema and sample cell values. SQLENS performs up to three correction iterations per query, resulting in a theoretical worst-case cost of 25,459 tokens. In practice, however, most queries are corrected within a single iteration, with an average token usage around 9,963 per query.

The overhead is minimal relative to state-of-the-art text-to-SQL systems. For example, CHASE-SQL [35] consumes approximately 160,000 tokens to generate a single SQL query. Manual SQL annotation by experts is even more costly, making SQLENS a practical and efficient solution for real-world use.

### 4.6 Analysis of Unfixed SQL Queries

Some SQL queries remain incorrect even after processing by SQLENS. To understand the causes, we conducted a detailed analysis and identified three main factors:

- Metric limitations. In some cases, the corrected SQL query produces the correct result but does not exactly match the provided ground-truth. For example, the fix may be semantically equivalent but use different columns or alternative formats, highlighting the limitation of the exact-match metric.
- LLM execution errors. The semantic errors are correctly detected, and SQLENS provides appropriate correction instructions. However, the LLM occasionally fails to follow these instructions accurately, preventing successful correction—an inherent limitation of the model itself.
- Incomplete error detection. In certain cases, the detected semantic errors do not fully capture the root cause. Consequently, even after fixing the identified issues, some errors remain in the query.

## 5 Conclusion

We present SQLENS, a novel framework to leverage both database- and LLM-based error signals for clause-level semantic error detection and correction in text-to-SQL. SQLENS uses weak supervision to aggregate noisy signals, predicts query correctness, and applies LLM-guided iterative correction. It outperforms LLM self-evaluation in error detection and improves execution accuracy of out-of-the-box text-to-SQL systems by up to 20% without relying on a correctness oracle.

## 6 Limitations

SQLENS leverages LLMs in SQL error detection and correction. Hence it inherits certain limitations from the underlying LLM. For example, context window constraints may restrict how much error information can be passed at once, and some corrections may exceed the LLM's current reasoning capabilities. In our analysis of unfixed SQL queries (Section 4.6), we observe cases where SQLENS correctly identifies semantic errors and provides appropriate guidance, but the LLM fails to apply the correction. SQLENS also incurs computational overhead from aggregating multiple signals and performing iterative corrections, which may introduce high latency in real-time settings. However, SQLENS is intended as an offline, general-purpose SQL debugging tool, designed to refine and validate generated queries asynchronously, rather than being deployed in latency-sensitive pipelines.

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

# A SQLENS Use Case

Figure 6 shows a text-to-SQL solution that generates a SQL query for a given NL question. Although the generated query has no syntax errors, the user executes the query and finds that it produces empty results, indicating the query is incorrect. Under the status quo in text-to-SQL assistants [4, 12, 41], users must invest significant time manually debugging semantic errors in queries. Our goal is to provide users with a `Debug SQL` functionality, which automatically performs a deep semantic analysis of the query, identifying errors such as incorrect join predicates in the `FROM` clause or incorrect usage of the provided external knowledge in the `WHERE` clause. This is followed by an automated correction process that generates an updated SQL query for the user to validate. Using `Debug SQL`, users' manual efforts in query troubleshooting are significantly reduced, consequently building trust in the text-to-SQL assistant.

# B DB-based Error Signals

The following signals detect semantic misalignment, which occurs when the LLM misinterprets the question or misuses the data.

• **Suboptimal join tree** signal targets the `FROM` and `JOIN` clauses, checking if the SQL query uses the optimal join tree to connect the required tables for answering the NL question. While query optimizers can eliminate redundant tables that do not affect the query results, LLMs may generate unnecessarily complex join trees, potentially leading to incorrect outcomes. Let $\mathcal{D}_{\text{req}} \subseteq \mathcal{D}$ denote the tables needed to answer the question. The optimal join tree is the minimum Steiner tree $T^*$ [19] spanning $\mathcal{D}_{\text{req}}$. If the query includes extra tables beyond $\mathcal{D}_{\text{req}}$, the *Suboptimal Join Tree* signal is flagged. For example, Figure 7 (right) shows an optimal join tree connecting tables $A$ and $C$. SQLENS identifies columns not involved in `JOIN` clauses as

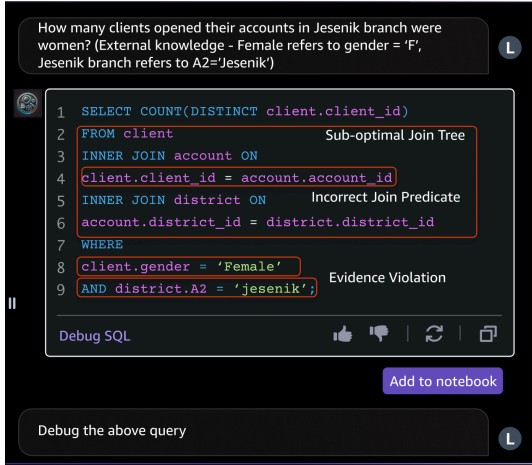

Figure 6: Text-to-SQL debugging use case.

$\mathcal{D}_{\text{req}}$ and uses Kruskal's algorithm [22] to find the minimum Steiner Tree on the pre-built join graph. If the query involves more tables than necessary, the signal is raised. False positives may occur when multiple valid join trees exist.

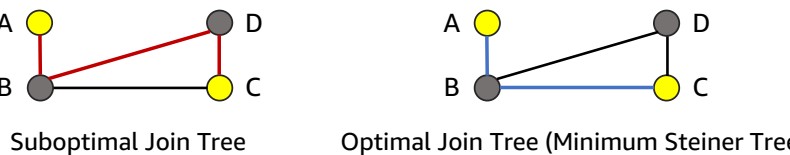

Figure 7: Steiner Trees spanning $A$ and $C$

To implement this signal, SQLENS first identifies the columns in a SQL query that are not involved in the JOIN clauses. Based on these columns, SQLENS searches for the minimum Steiner Tree on the join graph built offline to connect the relevant tables. If the SQL query involves more tables in the join than necessary, compared to the minimum Steiner Tree, SQLENS raises *Suboptimal Join Tree* flag. This signal can produce false positives when a suboptimal join strategy does not impact the correctness of the SQL query. For instance, if a query asks for the total sales from a specific store, it can directly select the *sales* column from the *store_sales* table. However, if the query unnecessarily joins the *store_sales* table with a *store_location* table, even though the *store_location* table is not needed to get the correct sales data, the result would still be correct but the join is suboptimal.

• **Incorrect join predicate** signal checks whether a SQL query uses an invalid join predicate. For example, in Figure 2, the predicted SQL incorrectly joins *client_id* with *account_id*, which does not

exist in the corresponding schema graph. To implement this signal, SQLENS first extracts all the join predicates from the SQL query. It then identifies two types of correct join predicates: (1) using a primary key-foreign key (PK/FK) join explicitly defined in the database schema, and (2) derived from PK/FK joins. Specifically, if two columns $C_i$ and $C_j$ both reference the same primary key $C_k$ (i.e., they refer to the same entity), $C_i$ and $C_j$ can be used in a join predicate. This signal may generate false positives when the PK/FK relationships are not fully documented in the database.

• **Empty predicate** signal detects if there is a predicate within a SQL query that yields an empty result. This signal is useful to detect semantic misalignment errors, including wrong column selection, wrong value usage or wrong comparison operator. SQLENS extracts all comparisons between a column and a literal from a given SQL, executes each of them individually and records the output size. If there is a predicate that yields empty rows, this signal is flagged. This signal may lead to false positives when an empty predicate is intentional.

• **Abnormal result** signal detects whether a SQL outputs an abnormal result that does not provide much information. SQLENS executes the SQL and records its output. The output is considered abnormal if (1) it is empty, (2) it contains a column full of zeros, or (3) it contains a column full of NULLs. This signal extends beyond the empty predicate to evaluate the entire SQL output. In addition to detecting empty predicates, it can identify empty intermediate execution results, making it effective for catching semantic misalignment in the intermediate steps of a SQL query. This signal may incorrectly flag a SQL query when the abnormal result is intentional, though this scenario is rare in practice.

• **Incorrect filter in subquery** signal detects the problematic filtering in a subquery. Filters in a subquery often follow the pattern *column* = (SELECT...). When the subquery returns multiple rows, the filter condition becomes ambiguous (e.g., IN or '='), potentially leading to errors. SQLENS uses regular expressions to match this pattern and executes the extracted subquery separately. If it returns more than one row, the signal is flagged. Additionally, SQLite is lenient with SQL semantics, allowing *column* = (SELECT...) to match only the first value returned by the subquery. While the query might still be correct if the first value happens to be the desired one, relying on this behavior is generally considered poor practice in SQL writing.

• **Incorrect GROUP BY** signal detects any standalone GROUP BY clause without accompanying aggregate functions such as MAX, COUNT. A misused GROUP BY clause can change the SQL semantics and lead to an incorrect result. In SQLite, a standalone GROUP BY behaves the same as the DISTINCT operator. While the query may still be correct when using GROUP BY as a substitute for DISTINCT, this approach is generally considered poor practice.

• **Unnecessary subquery** signal indicates if there is an excessive use of subqueries in a SQL query, which leads to inefficiencies, increased complexity, and a higher likelihood of errors. This signal counts the number of subqueries in a SQL query and flags it as problematic if the count exceeds a specified threshold. In our evaluation, this threshold is set to 3. False positives may arise when the subqueries are necessary for performance optimization or specific logic, even though they exceed the threshold.

The following signals are designed to capture the inherent ambiguity within the data.

• **Value ambiguity** signal detects incorrect column selections when a value used in an NL question appears in multiple columns. For example, "New York" can appear in both "state" and "city" columns. To identify ambiguous value, this signal extracts all values used in the SQL and finds columns that contain a used value via an inverted index built offline. If there is an alternative column that is closer to the question semantically, the signal flags the corresponding value as ambiguous. It is possible that the originally selected column in the SQL is correct, as the semantic distance may not always accurately determine which column containing the value is the best candidate to answer the question.

• **Table similarity** signal detects potential errors in table selection by identifying alternative tables with similar structures. It extracts all columns used in the SQL, groups columns by their tables, and searches for other tables that contain the same groups of columns. If such a table is found, it suggests that the alternative table could have been used, indicating a potential mistake in the original chosen table. False positives can occur when the chosen table is actually correct, but another table with a similar structure exists, leading the system to incorrectly flag a possible error.

Table 7: DB-based Error Signals.

| Signal Name | Incorrect SQL Query | Correct SQL Query | SQL Clause |
|---|---|---|---|
| Abnormal Result | SELECT c.id FROM cards c
WHERE c.cardKingdomFoilId = c.cardKingdomId
AND c.cardKingdomId IS NOT NULL
AND c.hasFoil = 1 AND c.isFullArt = 0
AND c.isOversized = 0 AND c.isPromo = 0; *Output Size: 0* | SELECT id FROM cards
WHERE cardKingdomFoilId IS NOT NULL
AND cardKingdomId IS NOT NULL;

*Output Size: 25061* | WHERE |
| Empty Predicate | SELECT c.atom_id, c.atom_id2 FROM connected c
JOIN bond b ON c.bond_id = b.bond_id
WHERE b.bond_id = 'TR_000_2_5'; | SELECT T.atom_id FROM connected AS T
WHERE T.bond_id = 'TR000_2_5'; | WHERE |
| Incorrect Filter in Subquery | SELECT Name FROM badges WHERE UserId =
( SELECT Id FROM users
WHERE DisplayName = 'Pierre' ); | SELECT Name FROM badges WHERE UserId IN
( SELECT Id FROM users
WHERE DisplayName = 'Pierre' ); | SUBQUERY |
| Incorrect GROUP BY | SELECT s.City, f.Enrollment (K-12)
FROM frpm f JOIN schools s
ON f.CDSCode = s.CDSCode
GROUP BY s.City, f.'Enrollment (K-12)'
ORDER BY SUM(f.'Enrollment (K-12)') ASC LIMIT 5; | SELECT T2.City FROM frpm AS T1
INNER JOIN schools AS T2
ON T1.CDSCode = T2.CDSCode
GROUP BY T2.City
ORDER BY SUM(T1.'Enrollment (K-12)') ASC LIMIT 5; | GROUP BY |
| Incorrect Join Predicate | SELECT (SELECT COUNT(DISTINCT client.client_id)
FROM client INNER JOIN account
ON client.client_id = account.account_id
INNER JOIN district
ON account.district_id = district.district_id
WHERE district.a2 = 'Jesenik'
AND client.gender = 'F') AS num_female_clients; | SELECT COUNT(T1.client_id)
FROM client AS T1 INNER JOIN district AS T2
ON T1.district_id = T2.district_id
WHERE T1.gender = 'F' AND T2.A2 = 'Jesenik'; | JOIN |
| Suboptimal Join Tree | SELECT d.a3 FROM client c
JOIN disp di ON c.client_id = di.client_id
JOIN account a ON di.account_id = a.account_id
JOIN district d ON a.district_id = d.district_id
WHERE c.client_id = 3541 LIMIT 1; | SELECT T1.a3 FROM district T1
INNER JOIN client T2
ON T1.district_id = T2.district_id
WHERE T2.client_id = 3541; | FROM
JOIN |
| Table Similarity | SELECT AVG(r.points) AS avg_score
FROM results r JOIN drivers d ON r.driverId = d.driverId
JOIN races ra ON r.raceId = ra.raceId
WHERE d.forename = 'Lewis' AND d.surname = 'Hamilton'
AND ra.raceId IN ( SELECT raceId FROM races
WHERE name LIKE '%Turkish Grand Prix%' ); | SELECT AVG(T2.points) FROM drivers AS T1
INNER JOIN driverStandings AS T2
ON T1.driverId = T2.driverId
INNER JOIN races AS T3 ON T3.raceId = T2.raceId
WHERE T1.forename = 'Lewis' AND T1.surname = 'Hamilton'
AND T3.name = 'Turkish Grand Prix' | SELECT
FROM |
| Unnecessary Subquery | SELECT (SELECT c.name FROM cards c WHERE c.uuid =
(SELECT uuid FROM rulings)) AS card_name,
(SELECT c.artist FROM cards c WHERE c.uuid =
(SELECT uuid FROM rulings)) AS artist,
(SELECT c.ispromo FROM cards c WHERE c.uuid =
(SELECT uuid FROM rulings)) AS is_promo; | SELECT T1.name, T1.artist, T1.isPromo
FROM cards AS T1 INNER JOIN rulings AS T2
ON T1.uuid = T2.uuid WHERE T1.isPromo = 1
GROUP BY T1.artist ORDER BY
COUNT(DISTINCT T1.uuid) DESC LIMIT 1 | SUBQUERY |
| Value Ambiguity | SELECT c.'artist' FROM 'cards' c
JOIN 'foreign_data' f ON c.'uuid'=f.'uuid'
WHERE c.'watermark'='phyrexian'
AND c.'artist' IS NOT NULL GROUP BY c.'artist'; | SELECT T1.artist FROM cards AS T1
INNER JOIN foreign_data AS T2
ON T1.uuid = T2.uuid
WHERE T2.language = 'Phyrexian'; | SELECT
WHERE |

In Table 7, we present examples of incorrect and correct SQL query pairs based on NL questions from real-world applications, highlighting the specific SQL clauses each signal targets.

# C   LLM-based Error Signals

• **Evidence violation** signal identifies cases where the generated SQL query contradicts the evidence provided in the question or external knowledge. For example, if the question specifies retrieving rows only about *active employees*, but the SQL query does not include a condition to filter out inactive employees, this would trigger an evidence violation. The prompt used by SQLENS can be found in Appendix D.2.

• **Insufficient evidence** signal assesses whether the available evidence is adequate to confirm that the SQL query correctly answers the user's question. For example, if the question lacks a clear explanation of a domain-specific concept, the LLM is prone to hallucination. This signal essentially verifies that the LLM has enough information and context to provide a correct response. The prompt is shown in Appendix D.3.

• **Incorrect question clause linking** signal evaluates the LLM's confidence in the generated SQL clauses. This signal first prompts the LLM to map the concepts, entities, and expressions in the user question to the corresponding clauses in the SQL query. For each identified link, the LLM is then asked to indicate its confidence in the generated clause by responding with a simple *yes* or *no*. This signal is flagged when there is at least one clause with low confidence. The prompt can be found in Appendix D.4.

• **Column ambiguity** signal identifies whether there are columns in the database that are very similar to those used in the SQL query and could also be used to answer the user's question. This signal

Table 8: LLM-based Error Signals.

| Signal Name | Incorrect SQL Query | Correct SQL Query | SQL Clause |
|---|---|---|---|
| Column Ambiguity | SELECT s.School, s.StreetAbr FROM satscores sat JOIN schools s ON sat.cds = s.CDSCode ORDER BY sat.AvgScrMath DESC LIMIT 1 OFFSET 5; | SELECT T2.MailStreet, T2.School FROM satscores AS T1 INNER JOIN schools AS T2 ON T1.cds = T2.CDSCode ORDER BY T1.AvgScrMath DESC LIMIT 5, 1; | SELECT |
| Evidence Violation | Evidence: set of cards with "Angel of Mercy" in it refers to name = 'Angel of Mercy'

SELECT COUNT(*) FROM set_translations WHERE setCode = 'UNH'; | SELECT COUNT(DISTINCT translation) FROM set_translations WHERE setCode IN ( SELECT setCode FROM cards WHERE name = 'Angel of Mercy' ) AND translation IS NOT NULL; | Any clause identified by LLM |
| Insufficient Evidence | Q: How many atoms with iodine and sulfur type elements are there in single bond molecules? Evidence: with iodine element refer to element = 'i'; with sulfur element refers to element = 's'; single type bond refers to bond_type = '-';

*It is not clear what "single bond molecules" refers to.* | SELECT COUNT(DISTINCT CASE WHEN T1.element = 'i' THEN T1.atom_id ELSE NULL END) AS iodine_nums, COUNT(DISTINCT CASE WHEN T1.element = 's' THEN T1.atom_id ELSE NULL END) AS sulfur_nums FROM atom AS T1 INNER JOIN connected AS T2 ON T1.atom_id = T2.atom_id INNER JOIN bond AS T3 ON T2.bond_id = T3.bond_id WHERE T3.bond_type = '-'; | Any clause identified by LLM |
| Incorrect Question Clause Linking | Q: List the top 10 players' names whose heights are above 180 in descending order of average heading accuracy. SELECT p.player_name FROM Player p JOIN Player_Attributes pa ON p.player_api_id = pa.player_api_id WHERE p.height > 180 ORDER BY pa.heading_accuracy DESC LIMIT 10; | SELECT t1.player_name FROM Player AS t1 INNER JOIN Player_Attributes AS t2 ON t1.player_api_id = t2.player_api_id WHERE t1.height > 180 GROUP BY t1.id ORDER BY CAST(SUM(t2.heading_accuracy) AS REAL) / COUNT(t2.'player_fifa_api_id') DESC LIMIT 10; | Any clause identified by LLM |
| LLM Self-Check | SELECT p.Paragraph_Text FROM Paragraphs p JOIN Documents d ON p.Document_ID = d.Document_ID WHERE d.Document_Name = 'Welcome to NY'; | SELECT T1.paragraph_text FROM Paragraphs AS T1 JOIN Documents AS T2 ON T1.document_id =T2.document_id WHERE T2.document_name = 'Customer reviews'; | Any clause identified by LLM |

suggests that a SQL is prone to wrong column selection. The prompt for this signal is shared in Appendix D.5.

The above signals provide specific error causes for the LLM to detect. Given that studies have shown LLMs possess the ability to validate their own answers [20, 45], we also incorporate a signal that prompts the LLM to provide an overall assessment of its own output.

• **LLM self-check** asks an LLM to determine whether the proposed SQL correctly answers the user question considering the database and any available external knowledge. The prompt can be found in Appendix D.6 and D.7.

The signal examples are listed in Table 8. False positives can occur for the LLM-based signals when the LLM misinterprets the question, has limited understanding of the specified knowledge, experiences hallucination, or exhibits bias when evaluating its own output.

# D   Prompts

## D.1   Prompt for Vanilla Text-to-SQL

```
Role: You are an expert SQL database administrator responsible for crafting
↪  precise SQL queries to address user questions.

Context: You are provided with the following information:
1. A SQLite database schema
2. A user question
3. Relevant evidence pertaining to the user's question

Database Schema:
- Consists of table descriptions
- Each table contains multiple column descriptions
- Frequent values for each column are provided

Your Task:
1. Carefully analyze the user question, evidence and the database schema.
2. Write a SQL query that correctly answers the user question

Format your SQL query using the following markdown:
```sql
YOUR SQL QUERY HERE
```
```

```
[Question]
{question}

[Evidence]
{evidence}

[Database Info]
{db_desc_str}

[Answer]
```

## D.2   Prompt for Evidence Violation

```
You are provided with a user question and a proposed SQL query that solves the
↪   user question. Your task is to determine whether the proposed SQL query
↪   reflects all the evidence specified in the question.

Here is a typical example:

==== Example ====

Question] How many users are awarded with more than 5 badges? more than 5 badges
↪   refers to Count (Name) > 5; user refers to UserId

[SQL query] SELECT UserId FROM ( SELECT UserId, COUNT(DISTINCT Name) AS num FROM
↪   badges GROUP BY UserId ) T WHERE T.num > 6;

[Answer]:

```json
{{
  "violates_evidence": true
  "explanation": "This SQL query violates the evidence in the question because it
  ↪   counts the number of users with more than 6 badges, not more than 5 badges.
  ↪   Additionally, it uses COUNT(DISTINCT Name) instead of COUNT(Name) as
  ↪   specified in the question.
}}
```
Question Solved. Make sure you generate a valid json response.
===============

Here is new example, please start answering:

[Question] {question}. {evidence}

[SQL query] {sql_query}

[Answer]
```

## D.3   Prompt for Insufficient Evidence

```
You are provided with sqlite database schema, a user question, and a proposed SQL
↪   query that solves the user question. Your task it to determine whether you
↪   have sufficient evidence to determine whether the SQL answers the user
↪   question correctly.

Output a JSON object in the following format. Make sure you generate a valid json
↪   response.
[Answer]
```json
{{
```

```
  "insufficient_evidence": true/false
  "explanation": "why the evidence is not sufficient"
}}
```
Question Solved.
===============
Here is new example, please start answering:

[Question] {question}. {evidence}

{db_desc_str}

[SQL query]

{sql_query}

[Answer]
```

## D.4 Prompt for Question-Clause Linking

```
You are given a SQLite database schema, a user question, and a proposed SQL query
↪  intended to address the user's question. Please follow these steps:

1. Link the concepts, entities, and expressions in the user question to the
↪  corresponding clauses in the SQL query.
2. For each link you have identified, indicate whether you are confident in the
↪  generated clause by answering "yes" or "no."

Output a JSON object in the following format. Make sure you generate a valid json
↪  response.
[Answer]
```json
{{
    "<(entity in the question, the corresponding SQL clause)>": "<yes/no>"
}}

```

Please start answering the following question:
[Question] {question}. {evidence}.

{db_desc_str}

[SQL query] {sql_query}

[Answer]
```

## D.5 Prompt for Column Ambiguity

```
As an experienced and professional database administrator, you are provided with a
↪  SQLite database schema, a user question, and a proposed SQL query intended to
↪  solve the user question. The database schema consists of table descriptions,
↪  each containing multiple column descriptions.
Your task is to determine whether there are columns in the database that are very
↪  similar to the ones used in the SQL query and could also be used to answer the
↪  user's question.

Here is a typical example:

==== Example ====
```

```
[Question] Which state special schools have the highest number of enrollees from
↪  grades 1 through 12? State Special Schools refers to DOC = 31; Grades 1
↪  through 12 means K-12

[DB_ID]california_schools
[Database Schema]
# Table: frpm
[
  (CDSCode, CDSCode.),
  (Enrollment (K-12), Enrollment (K-12).),
]
# Table: satscores
[
  (cds, cds. Column Description: California Department Schools),
  (sname, school name. Value examples: [None, 'Middle College High', 'John F.
  ↪  Kennedy High', 'Independence High', 'Foothill High', 'Washington High',
  ↪  'Redwood High'].),
  (enroll12, enrollment (1st-12nd grade).),
]
# Table: schools
[
  (CDSCode, CDSCode.),
  (DOC, District Ownership Code. Value examples: ['54', '52', '00', '56', '98',
  ↪  '02'].),
]

[SQL query] <SQL> SELECT T2.sname FROM schools AS T1 INNER JOIN satscores AS T2 ON
↪  T1.CDSCode = T2.cds WHERE T1.DOC = '31' AND T2.enroll12 IS NOT NULL ORDER BY
↪  T2.enroll12 DESC LIMIT 1; </SQL>

[Answer]
```json
{{
  "alternative_column": true
  "explanation": "frpm.Enrollment (K-12) can also be used to determine the number
  ↪  of enrollees from grades 1 through 12. This column is very similar to
  ↪  satscores.enroll12 used in the proposed SQL."
}}
```
Question Solved. Make sure you generate a valid json response.
===============

Please start answering the following question.
[Question] {question}. {evidence}

{db_desc_str}

[SQL query] {sql_query}

[Answer]
```

### D.6  Prompt for LLM Self-Check (True/False)

```
You are provided with a SQLite database schema, a user question, and a proposed
↪  SQL query intended to solve the user question. Your task is to determine
↪  whether the proposed SQL correctly answers the user question. Your answer
↪  should be in the form of a JSON object with two keys: "correct" and
↪  "explanation". Provide an explanation only if the SQL is incorrect.

You need to generate the answer in the following format:

[Answer]:
```json
```

```
{{
  "correct": false,
  "explanation": "your explanation of why the SQL is incorrect"
}}
```
Make sure you generate a valid json response.
================================
Please start answering the following question:
[Question]
{question}. {evidence}

[Database Info]
{db_desc_str}

[SQL query]
{sql_query}

[Answer]
```

### D.7 Prompt for LLM Self-Check (Probability)

```
You are provided with a user question, a SQLite database schema and a proposed SQL
↪   query intended to solve the user question.

Your task is to evaluate the proposed SQL query and provide the probability that
↪   it correctly answers the user question.

Provide this probability as a decimal number between 0 and 1.

You need to generate the answer in the following format:

[Answer]:
```json
{{
  "probability": <the probability between 0.0 and 1.0 that the SQL correctly
    ↪   answers the question>
}}
```
Make sure you generate a valid json response.
================================
Please start answering the following question:
[Question]
{question}. {evidence}

[Database Info]
{db_desc_str}

[SQL query]
{sql_query}

[Answer]
```

### D.8 Prompt for SQL Query Correction

```
Role: You are an experienced and professional database administrator tasked with
↪   analyzing and correcting SQL queries that are potentially wrong.

Context: You are provided with the following information:
1. A SQLite database schema
2. A user question
3. A proposed SQL query intended to answer the user question
```

```
4. An error report for the proposed SQL query. The error report suggests potential
↪  errors in the SQL.

Database Schema:
- Consists of table descriptions
- Each table contains multiple column descriptions
- Frequent values for each column are provided

Your Task:
1. Analyze the error report
2. Determine if the SQL query needs to be fixed. You can choose not to modify the
↪  SQL if it is correct.
3. If the proposed SQL is incorrect, generate a correct SQL query to answer the
↪  user question

Instructions:
1. Review the provided information carefully
2. Use SQL format in code blocks for any SQL queries
3. Explain your reasoning and any changes made to the query
4. Avoid using overly complex queries. For example, ... EXISTS (SELECT 1 FROM
↪  table WHERE condition) can be substituted with JOIN.

[Question]
{question}

[Evidence]
{evidence}

[Database Info]
{db_desc}

[Old SQL]
```sql
{old_sql}
```

[Error Report]
{error_report}

Now, please analyze the error report, decide whether the SQL needs to be fixed and
↪  generate a correct SQL to answer the user question if you think the proposed
↪  SQL is indeed wrong.

[Correct SQL]
```

## E SQLLENS Error Correction Pseudo Code

Algorithm 1 presents the pseudocode for the error correction. In essence, SQLLENS's error correction is a greedy, sequential algorithm. The original SQL query is first processed by SQLLENS's `Error Detector`, which evaluates a given set of signals on the input query and generates a list of error reports for the detected signals (Line 1). In practice, users can select a subset of signals to fix queries based on their precision in detecting errors on a development dataset. These reports are then sent to the `Error Selector`, which identifies the most critical error to address for the current iteration (Line 5).

The `Error Selector` component takes as inputs the original SQL query, database schema, and a list of error reports ranked by their confidence, and then determines the most critical error to address first using an LLM. Essentially, the LLM adjusts the ranking of these error reports based on its understanding of their relevance to the original SQL [43, 28, 37]. Once the most critical error is identified, the next step is to correct the query using the error information provided by that signal, handled by the `Error Fixer` (Line 6).

**Algorithm 1:** SQLENS Error Correction

---

**Input:** $ctx$: Correction context, including the database, external knowledge, and the natural language
question; $q$: Original SQL query; $\mathcal{S}$: Set of error signals; $max\_iter$: Maximum number of
iterations; $e_g$: Guardrail signal

**Output:** Corrected SQL query, $q'$

1   $\mathcal{E} \leftarrow ErrorDetector(ctx, q, \mathcal{S})$
2   $i \leftarrow 0$
3   $q' \leftarrow q$
4   **while** $|\mathcal{E}| > 0 \ or \ i < max\_iter$ **do**
        /* select the current most critical error                             */
5       $e \leftarrow ErrorSelector(ctx, \mathcal{E})$
6       $q' \leftarrow ErrorFixer(ctx, e, q')$
        /* remove error signal that has been fixed before             */
7       $\mathcal{S} \leftarrow \mathcal{S} \setminus \{e\}$
8       $\mathcal{E} \leftarrow ErrorDetector(ctx, q', \mathcal{S})$
9       $i \leftarrow i + 1$
10   **end**
    /* fix guardrail signal if detected                                   */
11   **if** $|ErrorDetector(ctx, \{e_g\})| = 1$ **then**
12       $q' \leftarrow ErrroFixer(ctx, e_g, q')$
13   **end**
14   $q' \leftarrow SQLAuditor(ctx, q', q)$
15   **return** $q'$

Table 9: Statistics of SQL queries generated by out-of-the-box text-to-SQL systems.

| Approach | Dataset | Acc. 1[1] | Acc. 2[2] | # SQLs | # Incorrect | # Semantic Err. |
|---|---|---|---|---|---|---|
| Vanilla | BIRD | 55.41 | 59.07 | 1534 | 684 | 589 |
| DINSQL | BIRD | 35.53 | 39.49 | 1534 | 989 | 835 |
| MACSQL | BIRD | 58.87 | 60.04 | 1534 | 631 | 601 |
| CHESS | BIRD | 67.60 | 67.91 | 1534 | 497 | 490 |
| Vanilla | Spider | 79.11 | 79.65 | 1034 | 216 | 209 |
| DINSQL | Spider | 76.31 | 77.66 | 1034 | 245 | 227 |
| MACSQL | Spider | 78.92 | 79.69 | 1034 | 218 | 208 |

[1] Accuracy over all queries [2] Accuracy over queries without syntax errors

The `Error Fixer` component takes the original SQL query, and a specific error report along with the correction context (specified in Algorithm 1) as input. It analyzes the error, determines how to fix the query, and generates a new query. To avoid syntax errors introduced in the revision process, we include a `Syntax Checker` in the `Error Fixer`, which uses a SQL parser to validate the syntax. If there is any syntax error, the `Error Fixer` uses the parser's error message to iteratively revise the SQL query until it is valid. Note that SQLENS avoids fixing the same error repeatedly by removing the error signal from the signal collection after it has been used for correction (Line 7). Once the current error is fixed, we run `Error Detector` on the revised SQL query. If it contains no further errors, the corrected SQL query is returned. Otherwise, the correction process iterates until all errors are resolved or the maximum number of iterations is reached (Lines 4-10).

The above process generates a revised SQL query, but it may still contain some errors that need to be fixed. To ensure the most critical error is corrected, we introduce a final check to identify if any further adjustments are needed. The signal with the highest precision in detecting incorrect SQL queries can be configured as a "*guardrail signal*" to catch and fix remaining errors (Lines 11-13). As mentioned earlier, the revision process may break an originally correct SQL query. To address this issue, the original SQL and the final revised SQL are passed to an LLM-based `SQL Auditor` for the final evaluation (Line 14).

# F    Additional Experimental Evaluation

Table 9 presents the text-to-SQL performance of the base SQL generators on BIRD and Spider.

## F.1 End-to-End Evaluation on Spider

Table 10 shows the end-to-end evaluation results on Spider benchmark. In terms of net improvement, SQLENS outperforms `Self-Reflection` by 10, 20, and 9 SQL queries on MAC-SQL, DIN-SQL, and Vanilla prompt, respectively. SQLENS consistently fixes more SQL queries compared to the baselines ($N_{\text{fix}}$). Its performance is similar to its variant, `Fix-ALL`, with SQLENS fixing 2 more SQL queries on MACSQL and nearly the same number on the other two methods. However, SQLENS produces more regressions than `Fix-ALL`. This is because error signals have lower accuracy on Spider than on BIRD, causing SQLENS to trigger more corrections, which results in a higher number of regressions.

Table 10: End-to-end accuracy improvement on Spider.

| Method | $\Delta$ Acc.($N_{\text{net}}$) | $\Delta$ Acc.($N_{\text{fix}}$) | $N_{\text{net}}$ | $N_{\text{fix}}$ | $N_{\text{break}}$ |
|---|---|---|---|---|---|
| **Vanilla** (Initial Acc.=79.65%); # input SQLs=1027 | | | | | |
| Self-Reflection | 80.14 (+0.49%) | 80.72 (+1.07%) | 5 | 11 | 6 |
| SQLENS w. Fix-ALL | **81.11 (+1.46%)** | 81.4 (+1.75%) | **15** | 18 | **3** |
| SQLENS | **81.11 (+1.46%)** | **81.5 (+1.85%)** | **15** | **19** | 4 |
| **DIN-SQL** (Initial Acc.=77.66%); # input SQLs=1016 | | | | | |
| Self-Reflection | 80.51 (+2.85%) | 81.1 (+3.44%) | 29 | 35 | 6 |
| SQLENS w. Fix-ALL | **82.58 (+4.92%)** | 82.88 (+5.22%) | **50** | 53 | **3** |
| SQLENS | 82.48 (+4.82%) | **83.27 (+5.61%)** | 49 | **57** | 8 |
| **MAC-SQL** (Initial Acc.=79.69%); # input SQLs=1024 | | | | | |
| Self-Reflection | 81.06 (+1.37%) | 81.45 (+1.76%) | 14 | 18 | 4 |
| SQLENS w. Fix-ALL | 81.74 (+2.05%) | 82.03 (+2.34%) | 21 | 24 | 3 |
| SQLENS | **81.93 (+2.24%)** | **82.23 (+2.54%)** | **23** | **26** | **3** |

## F.2 Error Detection on Spider

On Spider benchmark, we observe similar results, with SQLENS outperforming LLM self-evaluation methods in terms of both recall and F1 score (Table 11). A notable observation is that LLM self-evaluation tends to be overly confident in the generated SQL queries, leading to low recall in identifying incorrect queries. For instance, LLM Self-Eval. (Prob) only identifies 5.16% of the incorrect SQL queries on those generated by MAC-SQL. The overall F1 score on Spider is lower than on BIRD because the text-to-SQL algorithms already achieve around 80% accuracy on Spider, which is significantly higher than on BIRD. This creates a more challenging scenario, as most remaining errors fall within the long-tail distribution.

## F.3 Discussion

As confirmed by our experimental results, SQLENS is highly effective in error detection and correction, particularly benefiting non-technical users with limited SQL knowledge. However, like many other LLM-based text-to-SQL solutions, our method incurs some costs and latency. On average, SQLENS makes approximately 5 LLM calls with an end-to-end latency of less than 30 seconds. To further reduce costs and latency, users can configure cheaper error signals, provide ground truth labels for SQL correctness, or limit the number of iterations in SQLENS's error correction process. As noted in the motivating example in Section 1, SQLENS is not designed for real-time, interactive SQL query correction. Instead, it functions as a debugging tool, detecting and correcting errors in SQL queries asynchronously.

## F.4 Effectiveness of Individual Error Signals on BIRD and Spider

Table 11: Effectiveness of SQLENS error detection on Spider.

| Method | Acc. | AUC | Precision | Recall | F1 |
|---|---|---|---|---|---|
| Vanilla | | | | | |
| LLM Self-Eval. (Bool) | 74.39 (±3.12) | ✗ | 21.59 (±13.00) | 9.13 (±5.21) | 12.77 (±7.42) |
| LLM Self-Eval. (Prob) | **77.41** (±1.43) | 57.42 (±1.30) | 13.33 (±17.78) | 2.90 (±3.89) | 4.77 (±6.38) |
| Supervised SQLENS | **80.72**† (±1.06) | **61.02**† (±3.28) | **75.79**† (±17.24) | **10.56** (±4.53) | **17.79** (±6.62) |
| SQLENS | 74.49 (±4.86) | **59.49** (±5.91) | 34.91 (±14.18) | **29.22**† (±11.61) | **31.76**† (±12.75) |
| DIN-SQL | | | | | |
| LLM Self-Eval. (Bool) | 75.69 (±2.46) | ✗ | 40.38 (±11.33) | 16.78 (±4.24) | 23.64 (±6.01) |
| LLM Self-Eval. (Prob) | 76.58 (±1.23) | 58.60 (±1.78) | 41.07 (±17.97) | 5.73 (±2.23) | 9.90 (±3.72) |
| Supervised SQLENS | **82.48**† (±2.27) | **67.89**† (±3.66) | **73.68**† (±7.45) | **33.57** (±9.14) | **45.51** (±9.91) |
| SQLENS | 76.38 (±2.75) | **67.14** (±4.30) | 47.18 (±5.76) | **48.09**† (±6.91) | **47.59**† (±6.17) |
| MAC-SQL | | | | | |
| LLM Self-Eval. (Bool) | 76.85 (±1.60) | ✗ | 31.20 (±11.26) | **11.52** (±4.11) | **16.81** (±5.98) |
| LLM Self-Eval. (Prob) | 78.51 (±1.21) | 57.81 (±4.42) | **42.52** (±30.00) | 5.31 (±2.41) | 9.07 (±3.74) |
| Supervised SQLENS | **79.98**† (±1.39) | **61.59**† (±3.11) | **54.83**† (±21.06) | 9.65 (±8.11) | 15.10 (±11.29) |
| SQLENS | 74.41 (±2.24) | **58.99** (±3.19) | 35.88 (±5.22) | **32.23**† (±4.33) | **33.88**† (±4.49) |

✗ indicates that the classification is not threshold-based.
† Top result. Top two results are highlighted in bold.

Table 12: Individual error signal performance (Vanilla+BIRD).

| Signal Name | Precision | Recall | $N_w$ |
|---|---|---|---|
| **DB-based Signals** | | | |
| Abnormal Result | 98.96 | 16.13 | 95 |
| Empty Predicate | 85.34 | 11.88 | 70 |
| Incorrect Filter in Subquery | No Queries Detected | | |
| Incorrect GROUP BY | 40.0 | 2.38 | 14 |
| Incorrect Join Predicate | 100 | 1.87 | 11 |
| Suboptimal Join Tree | 45.9 | 14.26 | 84 |
| Table Similarity | 67.35 | 5.6 | 33 |
| Unnecessary Subquery | 33.33 | 0.34 | 2 |
| Value Ambiguity | 53.85 | 7.13 | 42 |
| **LLM-based Signals** | | | |
| Column Ambiguity | 68.75 | 1.87 | 11 |
| Evidence Violation | 61.11 | 1.87 | 11 |
| Insufficient Evidence | 29.03 | 1.53 | 9 |
| LLM Self Check | 60.82 | 10.02 | 59 |
| Question Clause Linking | 53.49 | 3.9 | 23 |

Table 13: Individual error signal performance (Vanilla+Spider).

| Signal Name | Precision | Recall | $N_w$ |
|---|---|---|---|
| **DB-based Signals** | | | |
| Abnormal Result | 60.0 | 14.35 | 30 |
| Empty Predicate | 50.0 | 2.39 | 5 |
| Incorrect Filter in Subquery | 100.0 | 0.48 | 1 |
| Incorrect GROUP BY | 64.29 | 4.31 | 9 |
| Incorrect Join Predicate | 100.0 | 5.26 | 11 |
| Suboptimal Join Tree | 42.86 | 5.74 | 12 |
| Table Similarity | No Queries Detected | | |
| Unnecessary Subquery | 100.0 | 0.48 | 1 |
| Value Ambiguity | 33.33 | 1.44 | 3 |
| **LLM-based Signals** | | | |
| Column Ambiguity | 33.33 | 0.48 | 1 |
| Evidence Violation | 100 | 1.44 | 3 |
| Insufficient Evidence | 7.14 | 1.44 | 3 |
| LLM Self Check | 20.65 | 9.09 | 19 |
| Question Clause Linking | 70.0 | 3.35 | 7 |

Table 14: Individual error signal performance (DIN-SQL+Spider).

| Signal Name | Precision | Recall | $N_w$ |
|---|---|---|---|
| **DB-based Signals** | | | |
| Abnormal Result | 73.61 | 23.35 | 53 |
| Empty Predicate | 82.98 | 17.18 | 39 |
| Incorrect Filter in Subquery | No Queries Detected | | |
| Incorrect GROUP BY | 43.33 | 5.73 | 13 |
| Incorrect Join Predicate | 92.86 | 11.45 | 26 |
| Suboptimal Join Tree | 33.33 | 5.73 | 13 |
| Table Similarity | No Queries Detected | | |
| Unnecessary Subquery | 0 | 0 | 0 |
| Value Ambiguity | 25.0 | 1.32 | 3 |
| **LLM-based Signals** | | | |
| Column Ambiguity | 80 | 1.76 | 4 |
| Evidence Violation | 80 | 1.76 | 4 |
| Insufficient Evidence | 40 | 7.93 | 18 |
| Question Clause Linking | 53.33 | 3.52 | 8 |
| LLM Self Check | 39.58 | 16.74 | 38 |

Table 15: Individual error signal performance (MAC-SQL+Spider).

| Signal Name | Precision | Recall | $N_w$ |
|---|---|---|---|
| **DB-based Signals** | | | |
| Abnormal Result | 58.82 | 14.42 | 30 |
| Empty Predicate | 52.94 | 4.33 | 9 |
| Incorrect Filter in Subquery | 100 | 1.44 | 3 |
| Incorrect GROUP BY | 50 | 3.37 | 7 |
| Incorrect Join Predicate | 100 | 3.85 | 8 |
| Suboptimal Join Tree | 38.71 | 5.77 | 12 |
| Table Similarity | No Queries Detected | | |
| Unnecessary Subquery | 33.33 | 0.48 | 1 |
| Value Ambiguity | 25.0 | 1.44 | 3 |
| **LLM-based Signals** | | | |
| Column Ambiguity | 0 | 0 | 0 |
| Evidence Violation | 58.33 | 3.37 | 7 |
| Insufficient Evidence | 18.92 | 3.37 | 7 |
| LLM Self Check | 31.17 | 11.54 | 24 |
| Question Clause Linking | 40 | 2.88 | 6 |

