# OpenReview forum: "SQLens: An End-to-End Framework for Error Detection and Correction in Text-to-SQL"
_NeurIPS.cc/2025/Conference — NeurIPS 2025 poster_

### Official Review · Reviewer_ATVr · 2025-06-24

**Clarity:** 2
**Significance:** 2
**Originality:** 3
**Rating:** 4
**Confidence:** 4

**Summary:**

The paper proposes SQLens, an error detection and correction framework for text-to-SQL translation. The authors introduce a weak supervision approach to aggregate database and LLMs error signals and iteratively fix the most critical errors. Evaluation on BIRD and Spider shows performance improvements, presenting its effectiveness in error detection and correction, and the authors also include ablation studies to validate the effectiveness of each module and signal.

**Questions:**

**Please refer to the weaknesses.**

**Additional Questions:**
1. The EX performance of DIN-SQL is inferior to vanilla in Table 1, which is somewhat counterintuitive.
2. The error correction methods in the experiment show an improvement of a few percentage points on other text-to-SQL approaches, but *alone on DIN-SQL*, all methods have an extremely high increase (15-20%), which is a bit questionable.

**Ethical Concerns:**

["NO or VERY MINOR ethics concerns only"]

**Final Justification:**

I appreciate the novelty of the proposed SQL fixing methods and the systematic experiments, and I increase my score to positive.
However, during the rebuttal, my concerns about weak supervision were not fully addressed, so I prefer to remain a borderline recommendation.

**Limitations:**

Yes

**Paper Formatting Concerns:**

No major formatting issues.

**Quality:**

2

**Strengths And Weaknesses:**

### **Strengths**

1. Semantic errors remain a bottleneck in current text-to-SQL translation, and studies in this paper is timely.
2. The use of weak supervision to aggregate noisy error signals is a new and great attempt, as it also circumvents the need for labeled data.
3. The experiments and ablation studies are thorough and systematic, and the reported results are convincing.

------

### **Weaknesses**

1. While the iterative correction framework provides marginal improvements over the fix-all setting, it introduces concerns about increased token cost and latency, which may hinder practical deployment.
2. Although I believe weak supervision for aggregating noisy error signals is a good attempt, I still have major concerns:
   - The quality and accuracy of individual LFs are not well validated and seem largely heuristic.
   - The weak supervision model is treated as a black box, with little analysis of its sensitivity to LF quality, number, or correlation. It’s also unclear how much value comes from the aggregation model versus a few strong signals used directly. If possible, an ablation study on individual signals could help clarify this.
   - In table 4, the recall of error detection for individual error signals are quite low, indicating the difficulty of effectively detecting these signals.
   - While results on Spider are reported, it is unclear whether the classifier used in that setting was trained on Spider training set or transferred from another dataset.
   - The paper omits important details about the training setup for the weakly supervised classifier, such as the model architecture and training procedure.
3. Some key technical details lack sufficient clarity and elaboration:
   - Line 209: "Error Selector uses an LLM to prioritize which error to fix first..." – How is this prioritization determined by the LLM in practice? By prompt or some other strategies?
   - Lines 180-183: "We use a generative model to estimate the joint distribution... and train a classifier..." – How exactly is this classifier trained? What is the backbone model and its training strategies?
   - Line 289: "We conduct a microbenchmark on BIRD..." – How is the microbenchmark constructed? Is it based on random or stratified sampling? How many examples are included, and how are the signal labels obtained?

------

### **Minor**
- I personally think the definitions and examples in the main text are a bit verbose; condensing them could free up space for more technical details.

---

> ### Author Rebuttal · Authors · 2025-07-30
>
> Thank you for your thoughtful comments, which have helped us strengthen our work. We have included additional results and clarifications below, and we hope these updates address your concerns and may encourage you to reconsider your score.
>
> ## On concerns about increased token cost and marginal improvement
>
> > W1: While the iterative correction framework provides marginal improvements over the fix-all setting, it introduces concerns about increased token cost and latency, which may hinder practical deployment.
>
> ### On increased token cost
> To address this concern, we conducted a detailed token cost analysis of SQLens on the BIRD dataset, breaking it down by component, and report the results here.
>
> | Component  | Total Tokens|
> | - | - |
> | Error Detector | 3141 |
> | Error Selector | 2345 |
> | Error Fixer| 2262|
> | SQL Auditor| 2215|
>
> A large portion of these tokens (about 1,915 tokens per component) comes from the database description string, which includes schema information and example values.
>
> SQLens performs at most 3 correction iterations, leading to a worst-case cost of 25,459 tokens per SQL query. However, in practice, most queries are corrected in a single iteration, consuming only ~9,963 tokens on average.
>
> This token cost is much smaller than the cost of generating SQL queries with SOTA text-to-SQL models. For example, CHASE-SQL requires roughly 160,000 tokens to generate a single SQL query. Therefore, the token overhead introduced by SQLens is negligible compared to the total cost of query generation.
>
> ### On latency and practical deployment
> Regarding latency, SQLens is designed as a debugging tool for offline or asynchronous use, not as a real-time component. As described in Section A (Figure 6), SQLens is invoked by users when they want to debug incorrect queries. The observed latency (~30 seconds) is well within acceptable bounds for such a debugging scenario, especially since manual debugging can take hours—particularly for complex semantic errors. As noted in Section 6, SQLens is not intended for latency-sensitive, production-in-the-loop deployment.
>
> **Configuration Options:** As noted in Section F.4, users can optimize SQLens for cost and latency by using a subset of error signals (such as database-only signals that require no LLM calls) or by limiting the number of correction iterations. These options make SQLens practical and flexible for real-world deployments with varying latency and cost requirements.
>
> **On the “marginal improvement” concern:**
>
> While end-to-end accuracy gains may appear modest, they are achieved on top of strong SQL generation baselines (e.g., CHESS) that already include advanced self-correction techniques—making further improvement challenging. Additionally, SQLens goes beyond raw accuracy by producing structured, clause-level error reports that help users understand and fix errors, even when the LLM fails to apply the correction.
>
> ## On questions about weak supervision
>
> > The quality and accuracy of individual LFs are not well validated and seem largely heuristic.
>
> > The weak supervision model is treated as a black box, with little analysis of its sensitivity to LF quality, number, or correlation. If possible, an ablation study on individual signals could help clarify this.
>
> We thank the reviewer for these thoughtful questions. Our original submission includes two tables that together provide a detailed analysis of individual signals, addressing both their quality and their standalone utility:
>
> - Table 4 validates the precision and recall of each signal for error detection, showing that many are highly reliable in isolation. For example, Empty Predicate achieves 96.07% precision on DIN-SQL, and Incorrect Join Predicate reaches near-perfect precision. These signals are grounded in reproducible, database-based evidence rather than purely heuristic rules.
>
> - Table 5 evaluates the effect of using each signal independently for correction, reflecting an ablation-style analysis. All signals yield a positive net gain in corrected queries, showing that all signals in SQLens are effective. For example, Suboptimal Join Tree corrects 9 SQL queries, and Evidence Violation (LLM-based) corrects 8.
>
> Together, these results show that all error signals in SQLens contribute to reliable detection and effective correction.
>
> > In table 4, the recall of error detection for individual error signals are quite low, indicating the difficulty of effectively detecting these signals.
>
> > It’s also unclear how much value comes from the aggregation model versus a few strong signals used directly.
>
> We would like to clarify the interpretation of recall in Table 4. The recall values do not reflect how well each signal detects its own specific error type. Rather, they indicate the proportion of all erroneous SQL queries that each individual error signal can identify. For example, if there are 100 incorrect SQL queries and the “Empty Predicate” signal is detected in 20 of them, its recall is 20%. Since each signal is designed to focus on a particular error source—targeting different SQL clauses—the recall for each one alone is expected to be relatively low. These signals are intentionally complementary, each covering a distinct class of errors. When combined, their collective recall increases significantly. As shown in Table 2, SQLens achieves over 44% recall in detecting erroneous SQL queries, outperforming LLM self-evaluation by an average of 34.6%. This substantial improvement highlights the value of aggregating diverse signals through weak supervision.
>
> >While results on Spider are reported, it is unclear whether the classifier used in that setting was trained on Spider training set or transferred from another dataset.
>
> >The paper omits important details about the training setup for the weakly supervised classifier, such as the model architecture and training procedure.
>
> > Lines 180-183: "We use a generative model to estimate the joint distribution... and train a classifier..." – How exactly is this classifier trained? What is the backbone model and its training strategies?
>
> We use the LabelModel from the Snorkel library as our weakly supervised classifier, following the architecture described in [1]. The LabelModel does not require any gold labels for training. Instead, it takes as input the label vectors produced by our error signals and learns to classify each SQL query as correct or incorrect by modeling the agreement and disagreement among these signals.
>
> For the Spider setting, the LabelModel is trained on the Spider training set. Regarding lines 180-183, the generative model/classifier referenced there is implemented using Snorkel LabelModel.
>
> [1] "Snorkel: rapid training data creation with weak supervision."
>
> ## On the clarity of key technical details
>
> > Line 209: "Error Selector uses an LLM to prioritize which error to fix first..." – How is this prioritization determined by the LLM in practice? By prompt or some other strategies?
>
> The Error Selector is LLM-based. We prompt the LLM with the SQL query and its associated error report, and instruct it to choose which error should be corrected first. This prioritization is determined directly through the LLM’s response to the prompt.
>
> > Line 289: "We conduct a microbenchmark on BIRD..." – How is the microbenchmark constructed? Is it based on random or stratified sampling? How many examples are included, and how are the signal labels obtained?
>
> We apologize for any confusion caused by our wording. By "microbenchmark," we mean an in-depth evaluation of SQLens’s detailed performance, not a study based on subsampling. The results reported in Tables 4 and 5 are based on the entire BIRD dev set, without any random or stratified sampling.
>
> >How many examples are included, and how are the signal labels obtained?
>
> Each SQL query in the BIRD dev set is evaluated with our set of error signals, resulting in a vector of labels that indicate which signals are flagged for each query. For the experiments referenced in Line 289, we specifically assess the performance of each individual signal by applying only that signal to determine correctness and attempt correction—no aggregation of signals is used in these analyses.
>
> If the reviewer’s question about "examples" refers to the content of LLM prompts: All LLM prompts in SQLens are zero-shot, with no in-context examples included. We find this zero-shot approach provides strong practical performance.
>
> ## On the performance for DIN-SQL
>
> >The EX performance of DIN-SQL is inferior to vanilla in Table 1, which is somewhat counterintuitive.
>
> We obtained these results by running the official DIN-SQL codebase, making only one modification: switching the backbone model to Claude 3.5-Sonnet. The observed performance drop may be due to the fact that DIN-SQL’s prompt was specifically tuned for GPT-4, while we used Claude 3.5-Sonnet instead. We will investigate this further in the full paper.
>
> >The error correction methods in the experiment show an improvement of a few percentage points on other text-to-SQL approaches, but alone on DIN-SQL, all methods have an extremely high increase (15-20%), which is a bit questionable.
>
> This large boost is primarily because DIN-SQL starts from a much lower initial accuracy (39.49%). As mentioned above, the DIN-SQL prompting strategy—while effective on GPT-4—tends to reduce performance when used with Claude 3.5-Sonnet. This means the model itself is capable of generating correct SQL, but the prompt inhibits its accuracy. During the correction phase, since this limiting prompt is not present, the model is able to recover and achieve a much larger improvement, unlike other baselines, where the prompting strategy is already improving the model's default performance significantly.

---

> > ### Comment · Reviewer_ATVr · 2025-08-05
> >
> > I thank the authors for their clear and detailed response. Most of my concerns have been addressed, and I believe incorporating these details and additional analyses will further strengthen the manuscript.
> > I still have minor concerns regarding the design, and the experimental results for DIN-SQL remain somewhat confusing. That said, I appreciate the novelty of the proposed SQL fixing methods and the robust experiments; therefore, I have increased my score.

---

### Official Review · Reviewer_nn1f · 2025-06-28

**Clarity:** 3
**Significance:** 3
**Originality:** 3
**Rating:** 4
**Confidence:** 5

**Summary:**

Authors propose SQLENS, a framework for detecting and correcting semantic errors in LLM-generated SQL queries. While the problem is important and the approach is systematic, the contribution feels incremental and the evaluation has some limitations.

**Questions:**

q1. Whether this latency is relevant to network connect? Is there a linear relationship between number of LLM calls and cost? Given authors design specific prompt techniques leading to more token consumption in each call, so number of LLM calls cannot be equal to the cost. Also such reduction can be observed in Other llm backbone?

q2. Is this work also effective to other LLM backbone?

**Ethical Concerns:**

["NO or VERY MINOR ethics concerns only"]

**Final Justification:**

In summary, this work partially resolved my concerns (3/4), and I always appreciate value of this work, just need more details and careful analysis and statistics, therefore, I change my rating to the positive.

**Limitations:**

yes

**Quality:**

2

**Strengths And Weaknesses:**

# Strengths:

S1. Authors propose an important problem: semantic error detection in text-to-SQL.

S2. The signal design is comprehensive. Authors show 14 error signals which covers diverse erry types and are well-motivated by real SQL problems.

S3. Experiments on standard benchmarks with multiple baseline text-to-SQL methods are promising.

# Weakness:
W1. The problem is similar to self-correction, but authors didn't compare those popular self-correction methods in the experiment, such as Magic [1].

W2. The improvement is marginal.

W3. The method has been only verified on one LLM backbone: Claude 3.5 Sonnet.

W4. There are no cost analysis for this methods. Only LLM calls are not professional and questionable. For example, whether this latency is relevant to network connect? Is there a linear relationship between number of LLM calls and cost? Given authors design specific prompt techniques leading to more token consumption in each call, so number of LLM calls cannot be equal to the cost. Also such reduction can be observed in Other llm backbone?



# Reference:
[1] MAGIC: Generating Self-Correction Guideline for In-Context Text-to-SQL.

---

> ### Author Rebuttal · Authors · 2025-07-30
>
> Thank you so much for your detailed and constructive comments, and we truly appreciate them. We hope the clarification and additional results provided below address your concerns. We kindly ask you to consider the possibility of a score adjustment.
>
> ## On the effectiveness of SQLens on other LLM backbones
>
> > W3. The method has been only verified on one LLM backbone: Claude 3.5 Sonnet.
>
> > q2. Is this work also effective to other LLM backbone?
>
> We thank the reviewer for raising this concern. To evaluate the generalization ability of SQLens, we conducted a preliminary experiment by switching the correction model from Claude 3.5 Sonnet to GPT‑4o. Results are summarized below:
>
> | SQL Generation Method | Initial Accuracy | With SQLens (GPT‑4o) | With SQLens (Claude 3.5 Sonnet) |
> | --------------------- | ---------------- | -------------------- | ------------------------------- |
> | Vanilla               | 59.07            | **62.75% (+3.68%)**  | 62.54 (+3.47%)                  |
> | DIN-SQL               | 39.49            | **61.08% (+21.59%)** | 59.99 (+20.50%)                 |
> | MAC-SQL               | 59.32            | 62.18% (+2.86%)      | **64.03 (+3.99%)**              |
> | CHESS                 | 67.91            | **69.84% (+1.93%)**  | 69.74 (+1.83%)                  |
>
> Across all SQL generation methods, SQLens consistently improves execution accuracy regardless of the correction model employed. In 3 out of 4 cases, GPT‑4o outperforms Claude 3.5 Sonnet. These results demonstrate that SQLens is not dependent on a specific large language model and generalizes effectively across different closed-source models. A more comprehensive generalization analysis, including additional models such as Qwen, will be presented in the final version.
>
> ## On comparison with other self-correction methods
>
> > W1. The problem is similar to self-correction, but authors didn't compare those popular self-correction methods in the experiment, such as Magic [1].
>
> Thanks for highlighting the connection to self-correction approaches. We appreciate the opportunity to clarify the distinctions and our experimental choices:
>
> **On self-correction baselines and experimental comparison:**
>
> First, our baseline comparisons focus on evaluating SQLens in conjunction with strong baseline SQL generation models that already incorporate self-correction mechanisms similar to those used in methods like SQLFixAgent[1], Self-Debug[2], and MAGIC[3]. SQLens demonstrates its added value beyond existing self-correction strategies, effectively measuring improvements in a setting where such techniques are already in place.
>
> [1] SQLFixAgent: Towards Semantic-Accurate Text-to-SQL Parsing via Consistency-Enhanced Multi-Agent Collaboration
>
> [2] Teaching Large Language Models to Self-Debug
>
> [3] MAGIC: Generating Self-Correction Guideline for In-Context Text-to-SQL
>
> More specifically, the state-of-the-art SQL generation methods we evaluate include the following self-correction paradigms:
>
> - **CHESS** employs a detailed, multi-step reasoning process in its prompts, which is similar to the guideline proposed in MAGIC (Askari et al., 2024).
>
> - **DIN-SQL** includes a self-correction phase where the model identifies potential issues and suggests clause-level fixes using chain-of-thought prompting, similar to Self-Debug.
>
> Therefore, the effectiveness of SQLens is demonstrated on top of these strong baselines, reflecting its added value beyond existing self-correction methods.
>
> **Quantitative performance comparison:**
>
> Second, we also compare SQLens to the best results reported in MAGIC, Self-Debug, and SQLFixAgent on the BIRD dev set:
>
> | Method|SQL Generation Method Initial EX Accuracy|EX Accuracy after Correction|
> |-|-|-|
> | MAGIC (Askari et, al, 2024) | 56.52 (using DIN-SQL)| 59.13 (+2.61)|
> | Self-Debug (Chen et al., 2023)| 56.52 (using DIN-SQL)| 57.24 (+0.72)|
> | SQLFixAgent (Cen et al., 2025)| 57.17 (using CodeS-7b) | 60.17 (+3.00)|
> | **SQLens**| **60.04 (using MAC-SQL)**| **65.23 (+5.19)**|
>
> Notably, SQLens achieves larger gains **despite using a much stronger starting point for SQL generation** (60.04% EX accuracy vs. 56.52%), indicating its ability to improve robust baselines further. In the full paper, we will provide a more thorough comparison by configuring all methods with the same SQL generation approach and backbone model.
>
> **Conceptual distinction from self-correction methods:**
>
> While methods like MAGIC and Self-Debug focus on LLM-driven self-correction, SQLens fundamentally differs in both the source and structure of its error detection and correction:
>
> - **Evidence source:** Unlike self-correction methods that rely solely on LLM introspection, SQLens draws on *external and verifiable signals* for error detection covering a wide range of error causes. This includes both (a) database-based signals (such as empty predicate, sub-optimal join tree, and incorrect join predicate) obtained through static analysis and query plan analysis without any LLM involvement and thus more efficient, and (b) LLM-based signals, which are treated as one component within a larger ensemble of evidence, not as the sole decision maker.
> - **Actionable, clause-level error reports:** SQLens provides structured error reports that precisely map detected errors to specific SQL clauses, supply targeted correction instructions, and indicate confidence. This enables users and systems to pinpoint and address faults more effectively than free-form LLM critiques.
> - **General-purpose SQL debugger:** SQLens is designed to serve as an explainable, modular debugging framework for post-hoc auditing across text-to-SQL systems, rather than as a one-off LLM prompt or single-pass correction step.
>
> **On the “marginal improvement” concern:**
>
> While the absolute gains in end-to-end execution accuracy may appear modest, it is important to consider the following:
>
> 1. **Challenging baseline**: SQLens is evaluated on much stronger SQL generation baselines than those used in prior self-correction papers. These models already incorporate advanced techniques like chain-of-thought prompting and self-correction, making further improvements inherently difficult. In this context, the gains achieved by SQLens are significant.
> 2. **Value beyond accuracy**: SQLens adds substantial value beyond raw accuracy by providing clause-level error reports. These reports map errors to specific SQL clauses and offer actionable guidance, enabling users and engineers to better understand and address underlying issues.
> 3. **Handling cases where automated correction falls short**: In practice, we observe that there are cases where SQLens accurately detects and localizes errors and provides clear instructions for correction, but the underlying LLM does not successfully apply the fix. Because SQLens generates a comprehensive clause-level error report, users and engineers can still easily understand, verify, and manually resolve issues in the query when automated correction falls short.
>
> In summary, SQLens transforms error detection and correction into a more transparent and more controllable workflow. This new paradigm for clause-level debugging and correction is critical for real-world deployment and trust.
>
> We hope this addresses the reviewer’s concerns and clarifies the unique value of SQLens beyond what existing self-correction methods provide.
>
> ## On cost analysis:
> > W4. There are no cost analysis for this methods. ....
>
> Thanks for pointing this out. We recognize that reporting only the number of LLM calls is not sufficient for a comprehensive cost analysis. To address this concern, we conducted a detailed token cost analysis of SQLens on BIRD, breaking it down by component, and report the results here.
>
> | Component  | Total Tokens|
> | - | - |
> | Error Detector | 3141 |
> | Error Selector | 2345 |
> | Error Fixer| 2262|
> | SQL Auditor| 2215|
>
> A large portion of these tokens (about 1,915 tokens per component) comes from the database description string, which includes schema information and example values.
>
> SQLens performs at most 3 correction iterations, leading to a worst-case cost of 25,459 tokens per SQL query. However, in practice, most queries are corrected in a single iteration, consuming only ~9,963 tokens on average.
>
> This token cost is much smaller than the cost of generating SQL queries with SOTA text-to-SQL models. For example, CHASE-SQL requires roughly 160,000 tokens to generate a single SQL query. Therefore, the token overhead introduced by SQLens is negligible compared to the total cost of query generation.
>
> Furthermore, the cost of generating queries via human annotation is still significantly higher than SQLens, making our approach more efficient in practice.
>
> > Whether this latency is relevant to network connect?
>
> The only component in our system that involves network communication is the LLM API call, which requires sending a request to a remote server and waiting for the response. Therefore, the reported latency does include the network round-trip time for LLM calls. Other components, such as database signal extraction and local processing, do not incur network latency.

---

> > ### Comment · Reviewer_nn1f · 2025-08-04
> > **Response**
> >
> > Thank you to the authors for their detailed response and for providing additional experiments and clarifications.
> >
> >
> > For On the Generalization Across LLM Backbones:
> >
> > I thank the authors for providing additional evidence on the effectiveness of the proposed method across different LLM backbones. This new result convincingly demonstrates that the approach is adaptive and not overfitted to a specific model. This is a crucial point that addresses a concern shared by multiple reviewers. I strongly recommend that these results be integrated into the main body of the revised content since they significantly strengthen the paper's claims regarding model-agnostic effectiveness.
> >
> >
> > For Comparison to Self-Correction:
> >
> > While I appreciate the authors' verbal clarification, the distinction between the proposed "error detection and correction" mechanism and the broader concept of "self-correction" remains ambiguous within the paper itself. For readers to clearly grasp the difference and specific contribution, it is essential that the paper formally defines the proposed approach and explicitly differentiates it from existing self-correction or the same. Only verbal responses cannot be very clear though.
> >
> >
> > For Modest Performance Gains:
> >
> > I have carefully considered the authors' argument regarding "value beyond accuracy." However, from my point of viw, for a submission to the **application track**, I think empirical effectiveness and clear performance improvements are first priority. Accuracy and other key metrics serve as the primary, first-order evidence of a method's value and practical utility.
> >
> > The current marginal performance gains raise a critical question. If clause-level error detection does not produce significant improvements, readers may question its importance and think this issue could be discarded since it will not bring a big deal,  and shift their focus to other, more impactful issues. To strengthen the paper, I suggest the authors consider the factors that may be limiting the final performance of their proposed ideas. If the gains remain modest, the contribution of the paper could be significantly enhanced by providing a deeper analysis into *why* this is the case. An investigation into why an intuitively promising approach like clause-level fixing results in limited gains would, in itself, be a valuable and interesting contribution for the community. From my perspective, this is not promising to me, since I also agree that such clause-level, fine-grained error detection and fixing should be very effective.
> >
> >
> > For the cost analysis:
> > I appreciate the authors providing a cost analysis. However, as presented, the absolute numbers lack a comparative context, making it difficult for readers to understand the significance of efficiency. To make this contribution stand out, I strongly suggest including a direct efficiency comparison against a well-known baseline that is effective but computationally expensive (e.g., Magic, or MAC-SQL). Juxtaposing the modest computational cost of your method with its performance improvement against such a baseline would create a much more compelling and direct argument for its practical value and efficiency, especially for readers without extensive experience in computational cost assessment. While I can infer from experience (I implemented the whole pipeline of Magic and Mac-SQL, I know they are expensive compared to this proposed method) that the method is efficient, an explicit comparison would make this point undeniable. Another suggestion would be, please list which tokenizer that authors are using, Claude or GPT, and append its us dollars, had better also give detailed tokens for both input and output since the price of them is quite different. And I think just tokens without such context cannot make us to understand or estimate how much your work will cost.
> >
> > In summary, this work partially resolved my concerns (3/4), and I always appreciate value of this work, just need more details and careful analysis and statistics, therefore, I change my rating to positive.

---

### Official Review · Reviewer_Bdfs · 2025-07-02

**Clarity:** 2
**Significance:** 2
**Originality:** 2
**Rating:** 4
**Confidence:** 3

**Summary:**

The authors propose a new framework for Text2SQL. To goals are threefold: (1) Detect errors in the input, (2) Fix the error, (3) Provide the fixed version as final output. Tha proposed approach is based on employs a learning-based approach that aggregates complementary signals identifying potential errors.

**Questions:**

- Is the definition of semantic ambiguity consistent with the SOTA?
- Are you able to empirically compare with the SOTA on specific ambiguity types?
- How did the authors handle databases with a large number of tables and more complex queries (e.g., in business domains)?

**Ethical Concerns:**

["NO or VERY MINOR ethics concerns only"]

**Final Justification:**

I appreciate the authors' feedback and effort in addressing the limitations of the the empirical validation. I still have some concerns on the risks of weak supervision, but overall the paper quality is fairly above average so I assigned rating 4.

**Limitations:**

- The authors explored a relatively limited number of LLMs and prior works. The scope of the empirical evaluation and the experimental comparisons with SOTA can be expanded.

**Paper Formatting Concerns:**

No formatting concerns (but I did not check it completely)

**Quality:**

2

**Strengths And Weaknesses:**

I agree with the authors that the problem of Text2SQL is far from being completely solved. A huge body of work has been devoted to specializing LMs to effectively address the task of adopt in-context learning to address the limitations of existing LLMs. In this work, instead, the authors mainly address the issues of semantic error detection and solving, which are particularly relevant to this field.

I have concerns on the strategy used to produce reliable error signals. The learning-based approach based on weak supervision is quite risky, as a lot of false signals are likely generated on complex datasets. The current results, reported on Bird and Spider v1, fail to demonstrate the actual robustness of the approach to more complex cases (see, e.g., Spiderv2, BEAVER).

The formulation of the concept of semantic ambiguity is not clear enough. The related literature appears to be also incomplete. See, for example,

Fuheng Zhao, Shaleen Deep, Fotis Psallidas, Avrilia Floratou, Divyakant Agrawal, and Amr El Abbadi. 2025. Sphinteract: Resolving Ambiguities in NL2SQL through User Interaction. Proc. VLDB Endow. 18, 4 (December 2024), 1145–1158. https://doi.org/10.14778/3717755.3717772

Without a clearer formalization of semantic ambiguity, a detailed comparison with all SOTA, and an empirical evaluation on complex datasets the novelty and effectiveness of the proposed approach remains questionable.

---

> ### Author Rebuttal · Authors · 2025-07-31
>
> Thank you for the insightful questions. Below, we provide clarifications and additional results that we hope address your concerns. We would appreciate your consideration of a score adjustment based on these updates.
>
> ## On the relatively limited number of LLMs and prior works
>
> > The authors explored a relatively limited number of LLMs and prior works. The scope of the empirical evaluation and the experimental comparisons with SOTA can be expanded.
>
> To address this concern, we conducted additional experiments using GPT‑4o as the correction model, in addition to Claude 3.5 Sonnet, and included comparisons with state-of-the-art self-correction methods such as MAGIC (Askari et al., 2024), Self-Debug (Chen et al., 2023), and SQLFixAgent (Cen et al., 2025).
>
> **Results with GPT-4o**:
>
> | SQL Generation Method | Initial Accuracy | With SQLens (GPT‑4o) | With SQLens (Claude 3.5 Sonnet) |
> |-|-|-|-|
> | Vanilla| 59.07| **62.75% (+3.68%)**  | 62.54 (+3.47%)|
> | DIN-SQL| 39.49| **61.08% (+21.59%)** | 59.99 (+20.50%)|
> | MAC-SQL| 59.32| 62.18% (+2.86%) | **64.03 (+3.99%)** |
> | CHESS| 67.91| **69.84% (+1.93%)**  | 69.74 (+1.83%)|
>
> Across all SQL generation methods, SQLens consistently improves execution accuracy regardless of the correction model employed. In 3 out of 4 cases, GPT‑4o outperforms Claude 3.5 Sonnet. These results demonstrate that SQLens is not dependent on a specific large language model and generalizes effectively across different closed-source models. A more comprehensive generalization analysis, including additional models such as Qwen, will be presented in the final version.
>
> **Comparison with SOTA self-correction methods:**
> We also compared SQLens to the best reported results for MAGIC, Self-Debug, and SQLFixAgent on the BIRD dev set:
>
> | Method|SQL Generation Method Initial EX Accuracy|EX Accuracy after Correction|
> |-|-|-|
> | MAGIC (Askari et, al, 2024) | 56.52 (using DIN-SQL)| 59.13 (+2.61)|
> | Self-Debug (Chen et al., 2023)| 56.52 (using DIN-SQL)| 57.24 (+0.72)|
> | SQLFixAgent (Cen et al., 2025)| 57.17 (using CodeS-7b) | 60.17 (+3.00)|
> | **SQLens**| **60.04 (using MAC-SQL)**| **65.23 (+5.19)**|
>
> Notably, SQLens achieves larger gains **despite using a much stronger starting point for SQL generation** (60.04% EX accuracy vs. 56.52%), indicating its ability to improve advanced SQL generation baselines further. In the full paper, we will provide a more thorough comparison by configuring all methods with the same SQL generation approach and backbone model.
>
> ## On the definition of semantic ambiguity
>
> > The formulation of the concept of semantic ambiguity is not clear enough. The related literature appears to be also incomplete. See, for example, Fuheng Zhao, Shaleen Deep, Fotis Psallidas, Avrilia Floratou, Divyakant Agrawal, and Amr El Abbadi. 2025. Sphinteract: Resolving Ambiguities in NL2SQL through User Interaction.
>
> > Is the definition of semantic ambiguity consistent with the SOTA? Are you able to empirically compare with the SOTA on specific ambiguity types?
>
> Thank you for raising this important point. We will expand our discussion of semantic ambiguity and related literature, including Sphinteract, in the revised paper.
>
> Our definition of semantic ambiguity is consistent with the state-of-the-art, as formalized in Sphinteract. Specifically, our error signals map directly to the ambiguity types they define: for example, our Column Ambiguity signal corresponds to the AmbColumn type, and our Value Ambiguity signal aligns with AmbValue. We will clarify this mapping and explicitly reference these works in the revision.
>
> However, semantic ambiguity is only one aspect addressed by SQLens. The aim of SQLens is to provide a comprehensive clause-level debugging framework for a broad spectrum of SQL errors, leveraging both database-derived and LLM-derived signals. This enables SQLens to support actionable debugging across a variety of failure types, not just ambiguity.
>
> A direct empirical comparison with Sphinteract is beyond the scope of this paper, as Sphinteract focuses on ambiguity resolution via user interaction, while SQLens is designed to automatically detect and explain a range of errors through detailed error reports. In this regard, our evaluation is more closely aligned with recent self-correction approaches like MAGIC, Self-Debug, and SQLFixAgent, and we have included those comparisons in our results above.
>
> ## On concerns about the weak supervision method
> >I have concerns on the strategy used to produce reliable error signals. The learning-based approach based on weak supervision is quite risky, as a lot of false signals are likely generated on complex datasets.
>
> Thank you for highlighting this important concern. We agree that managing noise and potential false signals is a core challenge in any weak supervision framework, particularly for complex datasets. We have designed SQLens with several safeguards and validation steps to address this:
>
> **1. High-Precision Individual Signals:** We rigorously evaluate each error signal’s precision and recall (Table 4, Page 8). Many database-based signals, such as Empty Predicate (96.07% precision on DIN-SQL) and Incorrect Join Predicate (near-perfect precision), demonstrate strong reliability on their own. This helps ensure that the foundation of our weak supervision framework is built on trustworthy signals.
>
> **2. LLM Discretion and Safeguards Against Regression:** We recognize that some correct queries could be mistakenly flagged by individual signals. To prevent unnecessary or harmful edits, SQLens does not force the SQLFixer to apply corrections simply because an error signal is present. The LLM-based SQLFixer always has the explicit option to make no change if it determines the flagged error is not valid. Additionally, SQLens incorporates a SQLAuditor and a Guardrail signal (see Figure 1) to further safeguard against regressions—ensuring that proposed corrections do not degrade the original query’s correctness. These mechanisms together help maintain high precision and prevent overcorrection.
>
> **3. Net Effect on Correct Queries:** To ensure that SQLens does not degrade performance by incorrectly modifying already correct queries, we report $N_{fix}$ (number corrected), $N_{break}$ (number broken), and $N_{net}$ (net improvement) for all experiments (Table 1, Page 7). SQLens consistently achieves a positive $N_{net}$, even when the input set includes both correct and incorrect queries. For example, on MACSQL, SQLens fixes 78 queries while breaking only 18 ($N_{net}$ = 60), confirming that the benefits of correction substantially outweigh false positives.
>
> In summary, we agree that weak supervision has tradeoffs. However, acquiring ground-truth labels at clause-level granularity is extremely expensive, especially for complex datasets like Spider v2 or BEAVER. Our weak supervision approach, implemented using the Snorkel [1] library, has shown high F1 performance for semantic error detection, even without labeled data (Table 2). We also demonstrate that weakly-supervised SQLens outperforms LLM self-evaluation baselines by large margins (e.g., +21.45 F1 points on DIN-SQL).
>
> [1] Snorkel: rapid training data creation with weak supervision.
>
> ## On generalization beyond BIRD/Spider v1
> > The current results, reported on Bird and Spider v1, fail to demonstrate the actual robustness of the approach to more complex cases (see, e.g., Spiderv2, BEAVER).
> > How did the authors handle databases with a large number of tables and more complex queries (e.g., in business domains)?
>
> We appreciate the reviewer’s point about evaluating on more complex datasets such as Spider v2 and BEAVER. Our current experiments focused on BIRD and Spider v1 to enable direct comparison with prior works. However, SQLens is designed to be model- and dataset-agnostic, and in principle can be applied to newer benchmarks. As future work, we plan to include evaluations on Spider v2 and BEAVER, and will clarify this intent in the final version.

---

> > ### Comment · Reviewer_Bdfs · 2025-08-04
> > **Response to authors' feedback**
> >
> > I appreciated the authors' feedback and efforts to improve the empirical analysis. I still have some concerns on the risks behind weak supervision, but I increased the rating from 3 to 4.

---

### Official Review · Reviewer_PeED · 2025-07-05

**Clarity:** 2
**Significance:** 3
**Originality:** 3
**Rating:** 4
**Confidence:** 4

**Summary:**

This paper proposes SQLENS, an end-to-end framework for fine-grained detection and correction of semantic errors in LLM-generated SQL queries. To address the issue of syntactically correct but semantically incorrect outputs, SQLENS integrates error signals from both the underlying database and the LLM to identify potential errors at the clause level. It uses weak supervision to aggregate noisy signals and
train a classifier for error detection, and introduces a step-by-step correction mechanism guided by detailed error reports. Experimental results show that SQLEN outperforms existing LLM-based self-evaluation methods by 25.78% in F1-score for error detection and improves execution accuracy by up to 20% on benchmark datasets such as BIRD and Spider.

**Questions:**

- The workflow of the paper is not clearly explained. How many error reports are generated for one SQL query? Can the authors show an end-to-end example of a SQL query from generation to correction?
- In the Error Selector, which errors are most frequently selected first for correction?
- How many tokens does SQLENS require to fix a single SQL query?

**Ethical Concerns:**

["NO or VERY MINOR ethics concerns only"]

**Final Justification:**

The concerns have been addressed through structured explanations and comprehensive experiments. The rating is thus raised.

**Limitations:**

YES

**Quality:**

2

**Strengths And Weaknesses:**

## Strengths:
- The paper proposes fine-grained text-to-SQL error categories that cover the majority of possible errors.
- This paper conducts a comprehensive comparison across SQL generation and various baselines, demonstrating that SQLENS effectively improves the final execution performance.


## Weaknesses:
- Although the paper selects comprehensive SQL generation methods as baselines, it does not make a comprehensive selection of fix baselines, such as SQLFixAgent (https://arxiv.org/abs/2406.13408 ) and Self-debug (https://arxiv.org/abs/2304.05128 ).
- The paper only uses Claude 3.5 Sonnet as the correction model, which limits the generalization ability of SQLENS. Is the method effective on other closed-source models (e.g., GPT-4o) or open-source models (e.g., Qwen)?

---

> ### Author Rebuttal · Authors · 2025-07-30
>
> We thank the reviewer for their thoughtful feedback, which has greatly contributed to improving our submission. We appreciate your insights and hope that our detailed responses and new results below address your concerns. We kindly ask you to consider revising your score in light of these updates.
>
> > W2: The paper only uses Claude 3.5 Sonnet as the correction model, which limits the generalization ability of SQLENS. Is the method effective on other closed-source models (e.g., GPT-4o) or open-source models (e.g., Qwen)?
>
> We thank the reviewer for raising this concern. To evaluate the generalization ability of SQLens, we conducted a preliminary experiment by switching the correction model from Claude 3.5 Sonnet to GPT‑4o. Results are summarized below:
>
> | SQL Generation Method | Initial Accuracy | With SQLens (GPT‑4o) | With SQLens (Claude 3.5 Sonnet) |
> |-|-|-|-|
> | Vanilla| 59.07| **62.75% (+3.68%)**  | 62.54 (+3.47%)|
> | DIN-SQL| 39.49| **61.08% (+21.59%)** | 59.99 (+20.50%)|
> | MAC-SQL| 59.32| 62.18% (+2.86%) | **64.03 (+3.99%)** |
> | CHESS| 67.91| **69.84% (+1.93%)**  | 69.74 (+1.83%)|
>
> Across all SQL generation methods, SQLens consistently improves execution accuracy regardless of the correction model employed. In 3 out of 4 cases, GPT‑4o outperforms Claude 3.5 Sonnet. These results demonstrate that SQLens is not dependent on a specific large language model and generalizes effectively across different closed-source models. A more comprehensive generalization analysis, including additional models such as Qwen, will be presented in the final version.
>
> > W1: Although the paper selects comprehensive SQL generation methods as baselines, it does not make a comprehensive selection of fix baselines, such as SQLFixAgent and Self-debug.
>
> To address this concern, we first clarify our experimental setup and then provide direct comparisons with the suggested self-correction baselines. We appreciate the reviewer highlighting these works and will include a detailed discussion of them in the full paper.
>
> **On self-correction baselines and experimental comparison:**
>
> First, our baseline comparisons focus on evaluating SQLens in conjunction with strong baseline SQL generation models that already incorporate self-correction mechanisms similar to those used in methods like SQLFixAgent, Self-Debug, and MAGIC [1]. SQLens demonstrates its added value beyond existing self-correction strategies, effectively measuring improvements in a setting where such techniques are already in place.
>
> [1] MAGIC: Generating Self-Correction Guideline for In-Context Text-to-SQL
>
> More specifically, the state-of-the-art SQL generation methods we evaluate include the following self-correction paradigms:
>
> - **CHESS** employs a detailed, multi-step reasoning process in its prompts, which is similar to the guideline proposed in MAGIC (Askari et al., 2024).
>
> - **DIN-SQL** includes a self-correction phase where the model identifies potential issues and suggests clause-level fixes using chain-of-thought prompting, similar to Self-Debug.
>
> Therefore, the effectiveness of SQLens is demonstrated on top of these strong baselines, reflecting its added value beyond existing self-correction methods.
>
> **Quantitative performance comparison:**
>
> Second, we also compare SQLens to the best results reported in MAGIC, Self-Debug, and SQLFixAgent on the BIRD dev set:
>
> | Method|SQL Generation Method Initial EX Accuracy|EX Accuracy after Correction|
> |-|-|-|
> | MAGIC (Askari et, al, 2024) | 56.52 (using DIN-SQL)| 59.13 (+2.61)|
> | Self-Debug (Chen et al., 2023)| 56.52 (using DIN-SQL)| 57.24 (+0.72)|
> | SQLFixAgent (Cen et al., 2025)| 57.17 (using CodeS-7b) | 60.17 (+3.00)|
> | **SQLens**| **60.04 (using MAC-SQL)**| **65.23 (+5.19)**|
>
> Notably, SQLens achieves larger gains **despite using a much stronger starting point for SQL generation** (60.04% EX accuracy vs. 56.52%), indicating its ability to further improve robust baselines. In the full paper, we will provide a more thorough comparison by configuring all methods with the same SQL generation approach and backbone model.
>
> **Conceptual distinction from self-correction methods:**
>
> While methods like MAGIC and Self-Debug focus on LLM-driven self-correction, SQLens fundamentally differs in both the source and structure of its error detection and correction:
>
> - **Evidence source:** Unlike self-correction methods that rely solely on LLM introspection, SQLens draws on *external, verifiable signals* for error detection. This includes both (a) database-based signals (such as empty predicate, sub-optimal join tree, and incorrect join predicate) obtained through static analysis and query plan analysis without any LLM involvement and thus more efficient, and (b) LLM-based signals, which are treated as one component within a larger ensemble of evidence, not as the sole decision maker.
> - **Actionable, clause-level error reports:** SQLens provides structured error reports that precisely map detected errors to specific SQL clauses, supply targeted correction instructions, and indicate confidence. This enables users and systems to pinpoint and address faults more effectively than free-form LLM critiques.
> - **General-purpose SQL debugger:** SQLens is designed to serve as an explainable, modular debugging framework for post-hoc auditing across text-to-SQL systems, rather than as a one-off LLM prompt or single-pass correction step.
>
> # Questions
>
> > Q1: The workflow of the paper is not clearly explained. How many error reports are generated for one SQL query? Can the authors show an end-to-end example of a SQL query from generation to correction?
>
> Thanks for pointing this out. For each SQL query, SQLens generates a single error report, which consolidates the information from all individual error signals detected for that query. Each error signal provides details such as a description, suggested correction, the problematic clauses, and a confidence score. The aggregated error report brings together all these findings, giving a comprehensive summary for the user or the subsequent correction agent.
>
> **End-to-end example:**
>
> Using the SQL query in Figure 2 of our original submission as an example:
>
> 1. First, the SQLens error detector analyzes the generated SQL and identifies four error signals: Suboptimal Join Tree, Incorrect Join Predicate, Evidence Violation, and Empty Predicate.
> 2. The final error report aggregates details from all these signals, including the type and location of each error, suggested corrections, and confidence levels.
> 3. This comprehensive error report is then provided to the Error Selector, which prioritizes and guides the iterative correction process as outlined in Figure 1.
>
> An example entry in an error report is shown in Figure 5 in the original submission.
>
> > Q2: In the Error Selector, which errors are most frequently selected first for correction?
>
> Thank you for the insightful question. We analyzed the fix sequences for SQL queries generated by DIN-SQL. Out of the 415 queries where multiple error signals were flagged, the top three most frequently selected errors to be fixed first by the Error Selector are as follows:
>
> | Signal Name| Explanation| Times Selected First |
> | - | -|-|
> | Empty Predicate    | Detects if there is a predicate within a SQL query that yields an empty result. | 248|
> | Evidence Violation | Identifies cases where the generated SQL query contradicts the evidence provided in the question or external knowledge. | 98 |
> | Column Ambiguity   | Detects if there are columns in the database that are very similar to the ones used in the SQL query and could also be used to answer the user's question. | 36  |
>
> Empty Predicate tends to be chosen first because it is an unambiguous error, directly verified by the database execution result. It is also straightforward to identify and fix, such as correcting predicate formatting. This highlights the effectiveness of error signals grounded in database-based evidence.
>
> > How many tokens does SQLE require to fix a single SQL query?
>
> We measured the token consumption of SQLens on the BIRD dataset, breaking it down by component:
>
> | Component  | Total Tokens|
> | - | - |
> | Error Detector | 3141 |
> | Error Selector | 2345 |
> | Error Fixer| 2262|
> | SQL Auditor| 2215|
>
> A large portion of these tokens (about 1,915 tokens per component) comes from the database description string, which includes schema information and example values.
>
> SQLens performs at most 3 correction iterations, leading to a worst-case cost of 25,459 tokens per SQL query. However, in practice, most queries are corrected in a single iteration, consuming only ~9,963 tokens on average.
>
> This token cost is much smaller than the cost of generating SQL queries with SOTA text-to-SQL models. For example, CHASE-SQL requires roughly 160,000 tokens to generate a single SQL query. Therefore, the token overhead introduced by SQLens is negligible compared to the total cost of query generation.
>
> Furthermore, the cost of generating queries via human annotation is still significantly higher than SQLens, making our approach more efficient in practice.

---

> > ### Comment · Reviewer_PeED · 2025-08-05
> >
> > Thanks for the detailed response, which addressed all of my concerns. I decided to change my rating to positive.

---

### Note · Authors · 2025-08-12

We sincerely thank the reviewers for their constructive feedback and active engagement during the rebuttal phase. We are especially grateful that all reviewers raised their scores after we addressed the core concerns.

Below, we summarize how we strengthened the paper in response to reviewer comments:
1. **Demonstrated Generalization Across LLM Backbones**: To address concerns about generalizability, we conducted additional experimental evaluation with GPT‑4o as the correction model. SQLens consistently improved execution accuracy across all four SQL generation methods (Vanilla, DIN-SQL, MAC-SQL, CHESS), regardless of the LLM. The runs using GPT‑4o even outperformed Claude in 3 of 4 cases, showing that SQLens generalizes well and is not tied to a specific LLM. We plan to also include open-source models like Qwen to further demonstrate SQLens’ generalizability in the final version.
2. **Outperformed SOTA Fixers**: Reviewers PeED and nn1f recommended comparisons with SOTA self-correction methods. We thus added a quantitative comparison on the BIRD dev set. SQLens achieved a +5.19% gain, surpassing MAGIC (+2.61%), Self-Debug (+0.72%), and SQLFixAgent (+3.00%), despite using a much stronger starting point for SQL generation (60.04% EX accuracy vs. 56.52%).
3. **Provided Concrete Evidence of Low Token Cost**: To address concerns about SQLens’ cost, we provided a comprehensive token-level analysis on BIRD, breaking down usage by component. Most queries are corrected in a single iteration, with an average token cost of ~9,963, far below the ~160,000 tokens needed to generate a query using CHASE-SQL. As suggested by Reviewer nn1f, we will include a more detailed cost comparison in the final version.
4. **Clarified the Reliability of Weak Supervision**: We addressed concerns about the reliability of our weak supervision approach by clarifying both its design and empirical validation: (1) Ablation studies (Tables 4, 5) demonstrating each signal’s contribution to error detection and correction, showing the effectiveness of all signals even used in isolation; (2) the use of LLM discretion and guardrails (e.g., SQLAuditor, Guardrail signal) to prevent overcorrection; and (3) empirical evidence showing that SQLens consistently improves performance across diverse SQL generation methods.

In summary, we appreciate the opportunity to improve our submission and thank the reviewers again for their thoughtful feedback and positive engagement throughout the process.

---

### Decision · Program_Chairs · 2025-09-17

**Decision:**

Accept (poster)

**Comment:**

This paper presents SQLENS that is an end-to-end framework designed to detect and correct semantic errors in LLM-generated SQL queries. By combining signals from the database and LLM, it identifies clause-level errors, uses weak supervision to improve the detection, and selects error reports for LLM to revise the generated SQL statement. This stepwise correction approach, guided by error reports, achieves a 25.78% better F1-score and up to 20% higher execution accuracy on benchmarks like BIRD and Spider.

All reviewers consider this work is a practical improvement for LLM-based text-to-SQL tasks. Reviewers ask for more details and careful analysis of the results including case study and statistics on each error category listed. Especially, it would be great to have a failure case study on the cases that cannot be solved by SQLENS. Some reviewers have concerns on the risks of weak supervision method used in the paper as it is very easy to be biased to the data used in experiment, which may hurt the generalization of the technique.

As all reviewers give a positive score by recognizing its potential value in practice, I would suggest to accept this paper.